# Climate pattern scaling set for an ensemble of 22 GCMs – adding uncertainty to the IMOGEN vn 2.0 impacts system

Przemyslaw Zelazowski[1,2], Chris Huntingford[3], Lina M Mercado[4,3] and Nathalie Schaller[1,5]

[1]Oxford University Centre for the Environment, University of Oxford, Oxford, OX1 3QY, UK

[2]Centre of New Technologies, University of Warsaw, Warsaw, 02-097, Poland

[3]Centre for Ecology and Hydrology, Wallingford, OX10 8BB, UK

[4]Geography, College of Life and Environmental Sciences, University of Exeter, Exeter, EX4 4RJ, UK

[5]Center for International Climate and Environmental Research (CICERO), Oslo, NO-0318, Norway

*Correspondence to*: Chris Huntingford (chg@ceh.ac.uk)

**Abstract.** Global Circulation Models (GCMs) are the best tool to understand climate change, as they attempt to represent all the important Earth system processes, including anthropogenic perturbation through fossil fuel burning. However, GCMs are computationally very expensive, which limits the number of simulations that can be made. Pattern-scaling is an emulation technique that takes advantage of the fact that local and seasonal changes in surface climate are often approximately linear in amount of warming over land and globe. This allows interpolation away from a limited number of available GCM simulations,

to assess alternative future emissions scenarios. In this paper, we present a pattern-scaling set consisting of spatial climate change patterns along with parameters for an energy balance model that calculates the amount of global warming. The set, available for download, is derived from 22 GCMs of the WCRP CMIP3 database, setting the basis for similar eventual pattern development for the CMIP5 and forthcoming CMIP6 ensemble. Critically, it extends the use of the IMOGEN (Integrated Model Of Global Effects of climatic aNomalies) framework to enable scanning across full uncertainty in GCMs for impacts

studies. Across models, the presented climate patterns represent consistent global mean trends, with maximum four (of twenty two) GCMs exhibiting opposite sign of the global trend per variable (relative humidity). The described new climate regimes are generally warmer, wetter (but with less snowfall), cloudier and windier, and with decreased relative humidity. Overall, when averaging individual performance across all variables, and without considering co-variance, the patterns explain one-third of regional change in decadal averages (mean Percentage Variance Explained, PVE, 34.25±5.21), but the signal in some

models exhibits much more linearity (e.g. MIROC3.2(hires):41.53) than in others (GISS_ER: 22.67). The two most often considered variables: near-surface temperature and precipitation have a PVE of 85.44±4.37 and 14.98±4.61, respectively. We also provide an example assessment of a terrestrial impact: changes in mean runoff, and compare projections by the IMOGEN system which has one land surface model, versus direct GCM outputs which all have alternative representations of land functioning. The latter is noted as an additional source of uncertainty. Finally, current and potential future applications of the

IMOGEN vn 2.0 modelling system in the areas of ecosystem modelling and climate change impact assessment are presented and discussed.

# 1 Introduction

Global Climate Models (GCMs) are the primary tool to understand and to estimate future climate regimes resulting from anthropogenic Greenhouse Gas emissions (GHGs). However, the use of these tools is limited by their requirements for computing power, and the complexity of the task, in particular when wishing to scan across different future scenarios. "Pattern-scaling" (Huntingford & Cox 2000, Mitchell et al 2003), is a methodology that takes advantage of the fact that, to a reasonable approximation, local and seasonal changes in surface climate are linear in amount of warming over land and globe. It allows interpolation away from a limited number of available GCM simulations, enabling a time-efficient assessment of surface meteorological changes for alternative non-standard future scenarios of changed GHG concentrations. This can include, for example, new scenarios that fall between the current Representative Concentration Pathways (RCP, Taylor et al., 2012), and potentially to investigate pre-defined future temperature thresholds such as two degrees of global warming above pre-industrial levels. Pattern-scaling has been suggested as a key methodology to understand the differences between and impacts of climate stabilisation at either 1.5°C or 2.0°C (James et al. 2017).

"Climate-change patterns" (or "patterns") are coefficients of the regression between areal mean warming over Earth's land regions, $\Delta T_1$ (K) and local changes in surface climatology. They are derived by comparison against outputs from GCMs, and presented as local monthly mean changes over land per degree of mean warming over land. "Pattern-scaling" is a simple procedure in which these patterns are multiplied by $\Delta T_1$ to give local monthly changes in climatology. A global Energy Balance Model (EBM; e.g. Wigley et al. 2000) is applied to model how the GHG concentrations translate in to changes in radiative forcing, $Q$ (Wm$^{-2}$) and then in to temperature increase over land regions ($\Delta T_1$).

The IMOGEN framework (Integrated Model Of Global Effects of climatic aNomalies; Huntingford et al. 2010) is a computationally efficient tool for modelling impacts of future climate change on terrestrial ecosystems. It consists of the JULES land surface model (Best et al., 2011; Clark et al., 2011) linked to a pattern-scaling module (Huntingford and Cox, 2000). The scaling provides monthly mean changes in climate variables over land, notably temperature, relative humidity, precipitation, shortwave and longwave radiation, wind speed and pressure – quantities used to drive ecosystem models such as JULES (Best et al., 2011; Clark et al., 2011). In addition, a simple oceanic global carbon cycle model is included (Joos et al., 1996 and Appendix of Huntingford et al., 2004), which expands the typical usage of pattern-scaling by allowing consideration of oceanic climate-carbon cycle feedbacks, alongside land-based feedbacks. All simulations use an hourly time step and a spatial resolution of 2.5° latitudinal x 3.75° longitudinal, or 72 by 96 grid boxes, as in the UKMO-HadCM3 GCM. Over land, but excluding Antarctica, this corresponds to 1631 grid-boxes.

Linking forcings to mean warming over land, $\Delta T_1$, is achieved with the IMOGEN EBM which requires five parameters (Huntingford and Cox (2000), also listed in Section 2.2) which, as in the case of patterns, are also derived by fitting to GCM runs. These calibration parameters and the previously-described patterns together form a "patterns-scaling set".

IMOGEN was originally established to allow rapid assessment of a range of alternative GHG emission scenarios, e.g. corresponding to any standard Special Report on Emissions Scenarios, (SRES, Nakićenović et al., 2000) or Representative

Concentration Pathways (RCP, Taylor et al., 2012) for which a GCM simulation is unavailable. Hence IMOGEN enables interpolation away from what GCM simulations do exist to new user-prescribed emissions or concentration pathways.

The general application of IMOGEN is to prescribe carbon dioxide ($CO_2$) emissions, along with further prescription of non-$CO_2$ radiative forcings for other GHGs and aerosols. From this, the model calculates evolving atmospheric $CO_2$ concentrations as a consequence of driving $CO_2$ emissions. The related and also evolving overall radiative forcing $Q$ ($Wm^{-2}$) is calculated to drive the EBM. This has similarities to how GCMs have been forced with the SRES scenarios in the third Coupled Model Intercomparison Project (CMIP3) archive, upon which this analysis is based. Alternatively, the full radiative forcing $Q$ can be prescribed directly as a future forcing pathway, dependent on prescription of all atmospheric gas changes and in which case $CO_2$ concentrations are therefore given. This has similarities to the more recent forcing of GCMs with RCPs, in part recognising that not all climate models have a fully interactive carbon cycle. RCPs were used to inform the 5[th] IPCC report (IPCC, 2013), via the set of climate model simulations available at that time in the CMIP5 dataset (Taylor et al 2012). CMIP5 has evolved further since year 2013 and to hold more simulations; the exercise to calibrate an EBM and patterns-scaling set for CMIP5 is under way. In addition, the scientific community is now starting to consider a broader range of scenarios, named Shared Socioeconomic Pathways (SSPs) (Riahi et al., 2017) and these will drive the forthcoming CMIP6 (Eyring et al., 2015) simulations.

In practice, though, IMOGEN has been used much more to assess the effects of new parameterisations, adjustment or inclusion of new processes into the JULES land surface model. This is as a precursor for any eventual placement of land surface model improvements in a full GCM. IMOGEN allows easy and fast assessment of ranges of parameterisations, numerical stability checks and critically the relative importance of new understanding of ecological and hydrological responses globally, and including feedbacks on the carbon cycle. Examples include the impacts of changes in diffuse radiation on the land carbon sink (Mercado et al., 2009), the effects of ozone damage on plant productivity (Sitch et al., 2007) and climate-carbon cycle feedbacks by permafrost melt (Burke et al., 2017).

Until recently, offline studies were performed with patterns of climate change from a single model, UKMO-HadCM3 (IMOGEN version 1.0, Huntingford et al., 2010). The purpose of this paper is to present a patterns-scaling set which emulates a broad range of GCMs, and nearly the complete set of those held in the WCRP CMIP3 database (Meehl et al., 2007). This extends the use of IMOGEN for assessment of climate change, or land surface response, to scan across uncertainty in both global response and local features of climate models. Such uncertainty can then be readily evaluated against the magnitude of any further uncertainty in any terrestrial surface impacts of interest. In Section 2 we describe the methods which lead to the patterns-scaling set. Section 3 describes the actual set, including discussion of inter-GCM differences. We include metrics describing the accuracy of the linearity assumption of meteorological changes against level of global warming, as implicit in the scaling method. Section 4 reviews existing applications of the IMOGEN pattern-scaling system and comments on the future general benefits of inclusion of climate model uncertainty in impacts assessments. Finally, Section 5 discusses the strengths and caveats of the pattern-scaling approach.

## 2. Data and Methods

### 2.1 The WCRP CMIP3 multi-model dataset and data pre-processing

Spatial patterns (i.e. maps) of climate change and energy balance model calibration parameters, together forming the "climate-patterns set", are derived from GCM data available through the World Climate Research Programme Coupled Model

Intercomparison Project, phase three (WCRP CMIP3; Meehl et al., 2007). The WCRP CMIP3 multi-model dataset resulted from an international effort to run a coordinated set of twentieth- and twenty-first-century climate GCM simulations for a limited number of future scenarios, covering many aspects of climate variability and change. All these simulations were subsequently analysed and formed the basis of much that is reported in the Fourth Assessment (AR4) of the Intergovernmental Panel on Climate Change (IPCC, 2007). The dataset consists of data from 24 GCMs, representing 17 modelling groups from

12 countries. The climate patterns set presented here (Table 1) corresponds to 22 GCMs, because GISS_AOM is an atmosphere-ocean model without surface meteorological projections over land and key data from CGCM3.1(T63) GCM were missing (see below). In the case of GISS-EH and GISS-ER GCM, WCRP CMIP3 data were supplemented with the formally associated pool provided by National Aeronautics and Space Administration Goddard Institute for Space Studies (http://data.giss.nasa.gov/pub/pcmdi).

The analysed model runs represent scenarios of four types: (i) control experiments: either pre-industrial or present day (Picntrl, or Pdcntrl; the codes are the file name conventions used in the WCRP database), (ii) the idealized 1% $yr^{-1}$ $CO_2$ increase up to doubled and quadrupled levels (1pctto2x, 1pctto4x), (iii) the twentieth-century run (20C3M) representing modelled period from pre-industrial to present-day and (iv) the high- (A2) and mid-emission (A1B) future scenarios defined by the Special Report on Emission Scenarios (SRES, Nakicenovic et al. 2000). When multiple simulations are available of any particular

scenario, then the analysis is limited to the first available, as the inter-run variability has been reported to be small (Frieler et al 2012).

Variables analysed for each GCM are those representing monthly mean land surface climatology: 1.5 m air temperature (TAS), 1.5 m relative humidity (HURS), 10 m wind speed (UAS and VAS, combined into a direction-less "UA"), precipitation (PR, including snow PRSN), downward shortwave (RSDS) and long-wave (RLDS) radiation fluxes and surface pressure (PS). The

codes in brackets are the file name conventions used in the WCRP database for individual variables. These seven variables are required to run the JULES land surface scheme inside the IMOGEN framework. Additionally, Net Radiative Flux at the top of the atmosphere (positive downwards); Top of Model (RTMT); is also processed. This is required to drive the global energy balance model.

There were some discrepancies between data requirements to run the IMOGEN system and the actual data availability in

WCRP CMIP3. They are listed in Table 1. For all GCMs, surface relative humidity (HURS) data were not available, but a 4D representation of this variable at pre-defined pressure levels (HUR) was generally available. This allowed extrapolation of surface relative humidity from two nearest available pressure levels. In the case of INGV-SXG, PCM and CCSM3 GCMs,

surface wind was obtained in the same procedure. For two cases, the required surface data was available, but suffered from quality and other issues. In UKMO-HadGEM1 dataset, the last month of the SRES A2 simulation was missing (and in this study it was filled-in with interpolated values), and surface wind data was presented on non-standard grid (and it was interpolated onto a standard UKMO-HadGEM1 grid). For MRI-CGCM2.3.2, many values in snow precipitation data (PRSN) were missing (the data were not used in this study) and there was no land mask available (SFTLF, later obtained from the Japanese modelling group). Additional details are given in Table 1.

GCMs differ significantly between each other in the spatial grid resolution and generally how they represent the Earth surface's detail, as represented in the land mask variable SFTLF which reports gridbox land fraction. Spatial resolution varies between hundreds of kilometres (e.g. GISS models, or INM-CM3.0) to around 100 km (e.g. MIROC3.2hires model, mid-latitudes, Table 1). Data are mapped on to either a regular or a Gaussian grid, and gridbox classification into land and water is either binary (100% or 0%) or continuous. Furthermore only some GCMs explicitly depict freshwater bodies in their land masks. This diversity of output spatial properties alone imposes a challenge for data end-users, including policymakers, especially when it comes to multi-model assessments of a pre-defined geographical domain. Hence, to force our common land surface model within the IMOGEN system using alternative GCM-based estimates of climate change, we harmonised all types of WCRP CMIP3 grids into one, which is chosen to be the UKMO-HadCM3, although land points for Antarctica are excluded. This ensures compatibility with previous applications of the IMOGEN tool, with resolution of 2.5° latitudinal x 3.75° longitudinal. The common grid allows, in a systematic way, to capture the impact of climate uncertainty that remains within GCMs. Details of the re-gridding procedure is provided in the Supplementary Material.

## 2.2 Climate pattern scaling set and post-processing

The presented climate patterns are a set of regression coefficients, each representing the change in a given meteorological variable per degree of mean global warming over land, while the fitting is done with decadal averages, as predicted by each GCM. The simple form of the analogue model for an anomaly (Δ) that is changing over decade and in one of the considered land surface variables $V$ is described as:

$$\Delta V(c, g, m, i) = \Delta T_l(c, i) V_x(g, m, i) \tag{1}$$

where the anomaly is linked to a single location on the UKMO-HadCM3 grid ($g$), month of the annual cycle ($m$), GCM ($i$) and decadal time index ($c$). Regressions to find (time-invariant) patterns $V_x$ use global land warmings $\Delta T_l$ directly from original GCMs to regress against. When the IMOGEN model is used predictively, then these values are derived using an Energy Balance Model (EBM) component, calibrated against different climate models (see below).

Regressing local climate with mean land warming is done with the assumption that climate is stable before the anthropogenic impact – often referred to as the pre-industrial period. This implies that the regression line starts at the origin of the coordinate system, so the intercept equals zero. Hence there is a fit with just one regression co-efficient; the slope fitted to modelled perturbed climate. This starting point is represented by an average of three decades from the twentieth-century run (20C3M,

years 1961-1990, Figure 1, panel C), recognising that this is later than pre-industrial. This is, however, a period corresponding to the Climate Research Unit's Time Series 2.1 (CRU TS 2.1) dataset describing Earth's climatology (or "climate normals", Mitchell and Jones 2005), and when that dataset is informed by a large number of widespread measurements. Patterns calculated by Eqn. (1) are generally added to the CRU dataset, rather than individual GCM estimates of pre-industrial times.

In the WCRP CMIP3 dataset, the historical 20C3M GCM simulations are normally followed by a future transient run, driven by one of SRES scenarios that describe potential pathways ahead in emissions. In the presented work, for most of GCMs, a high-emissions "business-as-usual" SRES A2 run was analysed to give patterns $\Delta V$ of Eqn (1). In a few cases when these data were not available the SRES A1B run was used (Table 2), which represents relatively lower levels of warming (Nakicenovic et al. 2000).

Emulating an ensemble of GCMs requires that the relationship between anthropogenic climate forcings, global warming and mean warming over land is established for each GCM separately. The EBM employed for this task, described in full in Huntingford and Cox (2000), requires the fitting of five calibration parameters: (i) an ocean effective thermal diffusivity, $\kappa$ (Wm$^{-1}$K$^{-1}$), (ii) a constant ratio of mean land and ocean surface (SST) rate of warming, $v$, (iii-iv) climate sensitivity over land $\lambda_l$ and ocean $\lambda_o$ (W m$^{-2}$ K$^{-1}$), and (v) land fraction $f$ (based on variable SFTLF, including Antarctica). All the energy retained

in the planetary system, as seen in any difference in top-of-the-atmosphere radiation, is assumed to enter the oceans in a diffusive process, thus changing SSTs and then $\Delta T_1$ via $v$. Estimation of EBM parameters was done by fitting them against an independent set of scenarios: the idealised $CO_2$ increase scenario (1pctto2x or 1pctto4x), preceded by a control experiment: (Picntrl or Pdcntrl). Subsequently, functioning of the parameterised EBM was validated by using it predictively against data from one of the available runs corresponding to SRES scenarios (SRES A2 or SRES A1B, Table 2). Figure 1 illustrates the

key components of the process of deriving a pattern-scaling set in the case of the example UKMO-HadGEM1 GCM.

In general, our climate patterns represent absolute changes. However, for precipitation, we make one additional calculation which results in data normalisation. This is to circumvent the problem of particularly large biases in the description of the current precipitation regime by some GCMs (Ines and Hansen 2006). For each calculated precipitation pattern ($\Delta P$), this is then multiplied by the ratio of the observed precipitation ($P_{\mathrm{CRU\_XXc}}$) from the CRU TS 2.1 dataset (Mitchell and Jones 2005)

and the one simulated by the GCM ($P_{\mathrm{GCM\_XXc}}$) for the control period. This follows Ines and Hansen (2006), and Malhi et al. (2009):

$$\Delta P'(g,m,i) = \Delta P(g,m,i) \times \frac{P_{\mathrm{CRU_{XXc}}}(i,m_S,g_S)}{P_{\mathrm{GCM_{XXc}}}(i,m_S,g_S)} \tag{2}$$

Furthermore, the adjustment described by Eqn. (2) was performed for each grid box $g$, month $m$ and GCM $i$, after smoothing in time and space (averaging over the grid box and its immediate neighbourhood: $g_S$, and across three months $m_S$). This

smoothing mitigates the minor shifts in seasonality and spatial positioning of climatic phenomena and reduces significantly the number of artefacts caused by occasional division by near zero. The remaining few cases of high and low divergence (i.e.

$P_{CRU,XXc}/P_{GCM,XXc}$) were capped at 5 and 0.2. Snow was scaled according to the same scaling factor as for total precipitation. The final patterns set is available in two versions: with precipitation normalised by Eqn. (2), and without this.

As a last step, in four cases when available GCMs data had one or two non-key variables missing (Table 1), the gaps were filled in with across-ensemble means.

## 3 Results

### 3.1 Energy balance parameters

The five key EBM parameters are presented in Table 2. In most cases (17) climate sensitivity over ocean ($\lambda_o$, Wm$^{-2}$K$^{-1}$) is higher than over the land ($\lambda_l$, Wm$^{-2}$K$^{-1}$). The reverse trend is well pronounced in three models (CSIRO-Mk3.5, MIROC3.2highres, MRI-CGCM2.3.2). Climate sensitivity over land varies five-fold between models and is the lowest in ECHO-G, UKMO-HadCM3, CGCM3.1(T47) and the highest in BCCR-BCM2.0, PCM, FGOALS-g1.0, although 2/3rds of the models have a much narrower range 0.9-1.7 Wm$^{-2}$K$^{-1}$. The most varying variable is ocean diffusivity ($\kappa$, Wm$^{-1}$K$^{-1}$), which determines the ability of the ocean to extract heat from the climate system through diffusion. Even after excluding the two most extreme cases, the range remains high: from 270 (HadCM3, CGCM3.1(T47)) to 2800 Wm$^{-1}$K$^{-1}$ (CSIRO-Mk3.0). The most extreme value of 11000 Wm$^{-1}$K$^{-1}$ is for the FGOALS-g1.0 GCM, which clearly stands out from the ensemble. (The FGOALS model has been noted as in outlier in other circumstance – e.g. Atlantic region projections; Perez et al., 2014). This spread reflects the fact that a full understanding of oceanic flows and deeper overturning, which affects mean vertical heat transport, is still required to reduce model spread. In comparison, the land/ocean temperature increase contrast ($\nu$) is a remarkably stable parameter, with a range 1.40-1.78 across all models.

### 3.2. Patterns across models, space and seasons

Across models, patterns of particular variables represent consistent trends when averaged spatially and across months (Table 3), with maximum four exceptions per variable (relative humidity), i.e. cases when average pattern is of opposite sign than in the majority of GCMs. The patterns capture the nature of a new emerging climate regimes, which can be characterised as warmer, wetter (but with less snowfall), cloudier and windier, with decreased relative humidity, and increased atmospheric pressure. Globally, relative humidity is the variable with the highest uncertainty in the magnitude of change, with standard deviation (SD) of the across-ensemble mean exceeding the mean. In the case of other variables, apart from longwave downward radiation and near surface air temperature (RLDS, TAS, with very small spread), the magnitude of SD is similar (SD of each variable is 62-88 % of the mean).

In the case of each GCM, the patterns represent a unique regional and seasonal distribution of change in surface climatology in a greenhouse-gas enriched atmosphere. To present these differences, we focus on two of the strongest drivers of terrestrial ecosystems change (and co-incidentally, which also have strong influence on society in general) – that is adjustments to

temperature and precipitation. The annual mean rate of warming per degree of global warming over land (Figure 2) in some models is much more evenly distributed geographically (e.g. BCCR-BCM2.0) than in others (e.g. NCAR-PCM1). However, all of the models exhibit the majority of warming in northern latitudes. The smallest warming occurs in tropical Africa and Asia, while in tropical South America the magnitude is much more uncertain. The spatial pattern of warming is either well

stratified with latitude (e.g. FGOALS-g1.0 model), or more nuanced (GISS models). The patterns of precipitation change (Figure 3) are more complex than in the case of temperature. The lack of across-ensemble consistency is particularly apparent in parts of the Tropics , e.g. in South America). However, in other areas the signal is very consistent, such as drying of Southern Europe, or more precipitation in high northern latitudes.

Across-model seasonal averages (Figure 4 and 5, for temperature and precipitation, respectively) reveal a more spatially and

temporally consistent picture than when considering models individually. These figures show that the majority of warming occurs at Northern latitudes and during colder seasons. Moreover, there is a strong summer warming trend over mid-West North America, Mediterranean region, Middle East and Central Asia. The seasonal patterns of precipitation change appear as linked to those of temperature, but are generally more uncertain. Winter warming in the North is accompanied by more precipitation which contrasts with lower summer warming and reduced rainfall. Changes in tropical rainfall appear as much

more uncertain. Western and Central Africa north of equator is a zone with particularly high uncertainty regarding summer warming.

Stippling in Figure 5 provides additional measure of uncertainty - it indicates when there is agreement in 90% of the models, as to whether precipitation is going to increase or decrease. This is the case over most of the land area and seasons. However, in many dry areas and seasons where this measure is not robust due to low precipitation levels (and the signal is difficult to

detect), the agreement is uncertain. Some areas stand out in this regard: large parts of South America in northern winter and summer, high northern latitudes in the summer and central Asia in autumn. That rainfall changes remain a large uncertainty in climate model projections is noted in the 4[th] IPCC report (IPCC 2007; Figure SPM.7) and in the 5[th] IPCC report (IPCC 2013; Figure TS.16).

### 3.3. Performance of linear approximation assumption in "pattern-scaling" for individual variables

The robustness of climate patterns is assessed by their ability to reproduce the decadal GCM data. Such ability varies widely between variables, which can be split into four categories, according to the mean Percentage Variance Explained (PVE) metric (Table 3). PVE is a simple way to assess each variable separately through the analysis of decadal means against the pattern. The most robust are the patterns which represent the drivers of global warming: temperature (TAS) and long-wave radiation (RLDS; PVE 85.44±4.37, and 84.74±4.97, respectively). The next group consists of variables which explain around one

quarter of variance: short-wave radiation (RSDS) and air pressure (PS). Variables linked to availability of water: precipitation (PR), snowfall (PRSN), relative humidity (HURS), form the third group (PVE 14.98±4.61, 17.96±4.67, 16.92±5.71, respectively). The last category is represented by wind patterns (of combined variables UAS and VAS), which represent only

7.11±3.32 PVE. Wind patterns also contain the highest proportion of negative PVE (4.9%) cases, for which a one-parameter regression line is a worse fit than a multi-decadal mean (bottom row of Table 3).

Overall (i.e. when per-variable results are averaged, without considering co-variance), climate patterns explain one-third of regional climate change (PVE 34.25±5.21); however, the signal in some models exhibits much more linearity (e.g. MIROC3.2(hires): 41.53) than in others (GISS_ER: 22.67). These estimates exclude cases where the PVE statistic could not be calculated due to either a lack of data (2.8%, Table 1), or null (e.g. short-wave radiation during polar night) or extremely low values (e.g. precipitation in the dry season), accounting for 6.7% of cases.

In terms of spatial distribution of robustness of the two key variables: temperature and precipitation (Figure 6), it is generally the opposite. For temperature, lower PVE values occur in the North, with minimum over Greenland and North-West North America (but still above 50%). The highest values occur across the Tropics. In the case of precipitation, the highest PVE occurs over the northern latitudes (above 50°N), particularly in Asia. In some tropical regions (sub-saharan Africa, South-East Amazon), areas with relatively robust signal (PVE ~20%) are adjacent to regions where the robustness could not be estimated due to very small and erratic rainfall in the dry season.

## 4. Applications

The "pattern-scaling" concept was originally designed as a tool for scientists to inform policymakers, enabling investigation of expected changes in surface climatology for a broader range of scenarios of atmospheric greenhouse gas concentrations than are available in archived GCM runs. The technique has been studied in depth, and one study recently concluded that "Overall, the well-established validity of the technique in approximating the forced signal of change under increasing concentrations of greenhouse gases is confirmed" (Tebaldi and Arblaster, 2014). The first version of the framework presented here was based around a single climate model. That is, to interpolate to new future greenhouse gas scenarios from the existing simulations by the UKMO-HadCM3 GCM (Huntingford and Cox, 2000). However, once the IMOGEN system (Huntingford et al., 2010) linked such scaling of meteorological drivers to force directly a land surface model (JULES), the main application of this system has been predominantly to undertake global analyses of new ecosystem process responses in a changing climate. This is often for similar forcing scenarios as GCMs have been operated for, but with the full climate models not yet running with the new land surface descriptions modelled in IMOGEN. Particular examples include quantification of wetland methane feedbacks (Gedney et al., 2004), the impacts of changes in diffuse radiation to the land carbon sink (Mercado et al., 2009), the effects of tropospheric ozone on plant productivity (Sitch et al., 2007), the significance of energy crop planting on future atmospheric $CO_2$ concentration (Hughes et al., 2010), how alternative mixtures of changes in atmospheric composition, even corresponding to identical radiative forcing changes, can have very contrasting impacts on land surface carbon stocks (Huntingford et al., 2011), and permafrost climate-carbon cycle feedbacks in a warming world (Burke et al., 2017).

The potential for Amazon forest collapse, or "die-back", remains an iconic concern for potential climate change impacts. Such a possibility has been identified in a combined climate-carbon cycle climate model UKMO-HadCM3LC (Cox et al., 2000, 2004). Later, the robustness of predictions of Amazon 'dieback' were investigated with IMOGEN (version 1.0) and the original UKMO-HadCM3 patterns (Huntingford et al. 2008), by analysing the vegetation response to (i) some limited uncertainty via

prescribed bounds in the parameterisations of the atmospheric component of UKMO-HadCM3 (related to HadCM3LC), (ii) description of canopy radiation interception – "big leaf" versus "multilayer" (Mercado et al., 2007) and (iii) representation of vegetation dynamics using an area based model and an individual based model (Moorcroft et al., 2001). All simulations show a fairly robust risk of dieback. More recently, a set of the climate patterns described in this paper were used to re-analyse the potential for tropical rainforest "die-back". Zelazowski et al. (2011) combined the patterns and global contemporary

climatology to produce high resolution maps of the future extent of humid tropical forests, while Huntingford et al. (2013c) forced the IMOGEN framework with the full set of patterns. Both studies found that climate models other than UKMO-HadCM3 are less likely to project such rainforest losses, which reflects the particularly strong climatic signal for the Amazon region temperature and precipitation changes for UKMO-HadCM3, as noted in Figures 2 and 3.

In order to exemplify the ability of IMOGEN to project changes to impacts, we report results of the mean annual total runoff

($R_{tot}$, mm day$^{-1}$) simulation based on the SRESA1B emissions forcing scenario (Figure 7), and compare them directly to GCM estimates of change in the same quantity. Hence this is comparing the IMOGEN simulations that emulate multiple GCMs but with a single land surface model (JULES), versus runoff values directly from the GCMs. The latter therefore contain alternative estimates of climate change, as well as the responses of different land surface models. During modelled pre-industrial control "spin-up" period, and modelled period centred on year 2090, total runoff values are recorded for each GCM (both emulated in

IMOGEN, and directly from GCMs). Then in each case, the change is calculated. In the top panels of Figure 7, the mean of these changes are shown, whilst the bottom panels are the standard deviations of these change values. Although there are similarities between left and right-hand panels (over northern latitudes, in particular), there are important differences too, and notably the drying signal in GCM output for Africa and Australia is not reflected in the IMOGEN framework. For SDs, in some locations there is higher variability for IMOGEN than for the GCMs themselves; however, this pertains mainly to the

regions where IMOGEN predicts higher runoff. The latter may be surprising, as considering that GCMs directly introduce another level of uncertainty i.e. inter-land surface model differences. Our finding is suggestive that JULES has a particularly sensitive response of runoff to imposed climatic changes. Runoff provides a challenge for comparison, as it is frequently a relatively small number between two larger fluxes of precipitation and evapotranspiration (transpiration, plus soil evaporation and interception loses) and so sensitive to change in those fluxes. Any direct comparison also needs to account for IMOGEN

being initialised with a climatology based on the CRU dataset, and temporal dis-aggregation to sub-daily drivers of JULES having not been calibrated against any particular GCM. Nevertheless, to be a useful tool for impacts assessment, then IMOGEN must capture the general features of GCM projections of quantities such as runoff when operated for similar emissions scenarios.

Looking ahead to further model development, one possibility is, for different GCMs, to pattern-scale directly impact variables of interest such as runoff against global land temperature change. This could be beneficial for two reasons. First, it would remove the current IMOGEN mismatch of many GCMs emulated based on their climate projections only, while the emulating system is coupled to just one land surface model. Instead, a more accurate representation would be gained of the spread of

runoff uncertainty. Second, it would make model calculations computationally very fast, as full operation of the JULES system would not be required. Such an approach would be applicable when using IMOGEN to estimate changes for different future greenhouse gas concentrations, rather than land surface modelling development. A further possibility is to connect the meteorological pattern-scaling structure to alternative land surface models. The analysis of Sitch et al. (2008) used the IMOGEN system to diagnose uncertainty in representation of future plant biogeography and climate-carbon cycle feedbacks

using five Dynamical Global Vegetation Models (DGVMs), but then combined with only a single set of climate model patterns based on UKMO-HadCM3. This could be re-visited. If each DGVM modelling centre could operate their latest DGVM configuration, across the range of emulated GCMs, then this would give a fuller estimate of the balance between implications of uncertainty in climate and uncertainty in terrestrial ecosystem response and its feedbacks on the global carbon cycle. A full set of calculations would entail 5 times 22 simulations, for a single future atmospheric greenhouse gas concentration pathway

or emissions pathway. This capability could be of interest to research programs designed to compare different estimates of impacts under climate change (e.g. the Inter-Sectoral Impact Model Intercomparison Project, ISIMIP).

In some regards, land surface models in GCMs are still in their infancy, considering the growing knowledge of how vegetation responds physiologically to imposed climatic changes. For this reason, there are future plans to use IMOGEN as an intermediate step, before inclusion in a full GCM, to test and demonstrate the relative importance of new process descriptions.

For example, we have used the patterns derived in this paper on an analysis of the sensitivity of the future land carbon storage to thermal acclimation of plant photosynthesis (Mercado et al., in preparation). This is a noted major deficiency in current large-scale terrestrial models (Booth et al 2012; Huntingford et al., 2013b; Smith and Dukes 2013). In addition, the assessment of newly available enhanced description of leaf dark respiration (Atkin et al., 2015) is needed, as well as the inclusion of both Nitrogen and Phosphorus constraints to plant productivity in tropical ecosystems (e.g. Mercado et al., 2011), and inclusion of

a full representation of a coupled Carbon-Nitrogen cycle in JULES (Zaehle et al 2010). Furthermore, it is desirable to test the effects of adding height competition into the vegetation dynamics module of JULES, in order to add ecological succession modelling (Smith et al 2001; Moorecroft et al 2001), along with assessing the impacts of improved representation of stomatal conductance (Medlyn et al 2011; Kala et al 2015) and plant hydraulics (Sperry et al. 2015) on simulated land carbon and water cycles couplings to climate. The latter could extend as far as testing any hormonal signalling in the hydraulic linkages between

soil moisture and stomata response; an effect well known by the physiological community but hereto never tested in a full large-scale gridded land surface model (Huntingford et al., 2015). Finally, impacts of introducing a better representation of plant functional types and plant trait variation across space and time (Verheijen et al 2015) on simulated land carbon could also be considered.

## 5. Discussion

In this paper we present a "pattern scaling set", consisting of spatially explicit climate change patterns and EBM calibration parameters, which together represent 22 GCMs of the WCRP CMIP3 database (Meehl et al., 2007). This data set extends the use of the IMOGEN climate impacts assessment tool to scan across uncertainty in climate models. Despite relying on a set of simple assumptions, the tool can capture a significant part of the predicted changes in surface climatology. Terrestrial ecosystem response studies have used this modelling framework to gain new insights into how the land surface component of the Earth system functions. A new version of the pattern scaling set, based on the CMIP5 dataset, will build on those available here.

The IMOGEN modelling system is available to determine future climate change, now with uncertainty, and forced by either a future pathway in either $CO_2$ emissions or $CO_2$ concentrations. A further and rapidly emerging application is to understand regional climate impacts during transition to different global thermal limits, with an emphasis on eventual stabilisation at 1.5°C or 2.0°C of global warming above pre-industrial levels. In this instance, most of the EBM in IMOGEN is overridden with a global temperature pathway (the land-ocean contrast and oceanic fraction cover only used to obtain $\Delta T_1$), but relying on the remaining spatial and monthly patterns to give detailed local climatic implications. It is planned to use different global temperature pathways to those two stabilised limits (Huntingford et al, 2017) to force IMOGEN in this configuration. A further advantage of the IMOGEN system is that the projected anomalies of meteorological change are generally added to known gridded climatologies, rather than to any GCM-based baseline estimates. One such dataset is the routinely updated Climate Research Unit (CRU) climatology (Harris et al., 2014). A typical estimate of pre-industrial conditions may be regarded as the mean climatology of a period such as 1960-1989, a time when weather stations had become much more available worldwide to guide the dataset construction. Although this will fail to capture warming effects between pre-industrial times and that period, such offsets may be much smaller than the biases removed by not using GCMs to estimate a baseline climatology to which IMOGEN anomalies are added. This does, though, assume such GCM bias removal is valid for the entirety of any transient simulation. Recent analysis appeals for more process information to be accounted for when attempting bias correction (Maraun et al, 2017).

An important aspect of the presented work is the comprehensive study of the pattern's robustness, i.e. their ability to capture variability of climate simulated by GCMs. That across all variables considered, 1/3rd of decadal variability in monthly averages is captured, suggests that it is a technique with a significant potential. This is especially since it allows a large reduction in input data and computation requirements compared to full GCMs. Overall, the presented patterns are in good agreement with the results presented in the Fourth Assessment Report of the Intergovernmental Panel on Climate Change (e.g. for precipitation SI Fig. 2 and Figure 10.9 in Meehl et al., 2007). This applies to both the multi-model mean changes in surface climate, as well as the degree of agreement between the models (stippling in Figure 5).

However, the ability of climate patterns to capture the course of changes varies significantly between the modelled climate variables. In contrast to temperature change (85.44±4.37% of variability explained by climate patterns), change in precipitation

is usually more difficult to capture in this linear methodology. Overall 85% of precipitation variability remains unexplained, although it should be noted that in some regions seasonal precipitation patterns explain up to 75% of variability (generally at high northern latitudes). The relatively weak explanatory power of the precipitation patterns can be partly explained by poor trend estimation over dry zones.. In these areas, the mean change over decades gives low PVE values, as any change in very

small current absolute precipitation can result in high relative deviations from preindustrial levels. In such circumstances, precipitation is very much dominated by inter-annual fluctuations.

In addition to uncertainty linked to methods and assumptions (such as linearity), there is some uncertainty linked directly to driving and calibrating input data. The decision to use 20C3M and SRES scenarios to derive patterns, and the idealised scenario of 1% annual $CO_2$ increase to calibrate the EBM model, reflects a compromise between the accuracy of patterns and forcings.

It could be argued that both the patterns and the EBM parameters should be derived from the same set of GCM runs. However, since the SRES runs are on average longer (12 decades, with part of the 20C3M run), therefore they are a better source for linear fitting of the spatial patterns (Mitchel 2003). However the idealised $CO_2$ increase scenarios are a better basis for energy balance model calibration as the definition of SRES forcings varies between modelling groups (they often encompass atmospheric aerosols) and are less well documented.

Although placing climate data on a common grid brings strong benefits to the IMOGEN tool, there is also a compromise. For the re-gridding method combined with land mapping (see Supplementary Material), the calculated regional patterns represent areas that are comprised fully of land, while in much of original GCM data grid-boxes represents a mix of land and ocean. The total land fraction in the presented spatial patterns is slightly increased due to this (see Figure S1). This increases the average grid-box warming, due to diminished representation of the oceanic heat uptake. As a result, the fitting procedure yields regional

warming patterns (column "TAS" in Table 3) which, when area-weighted, overall return a value slightly larger than 1.0 K/K. However, this effect has no impact on the global energy budget in the IMOGEN framework, which is resolved independently with EBM.

There are a number of potential methodological enhancements that can be implemented in the next IMOGEN version and beyond just fitting to the CMIP5 (Taylor et al., 2012) or CMIP6 (Eyring et al., 2016) datasets. For example, so far the natural

variability around the trend in IMOGEN is simulated through a daily "weather generator" component with invariant properties, and with no representation of inter-annual variability. However variability might also change in a warming world, and at a range of timescales from sub-daily through to major alteration at inter-annual timescales (e.g. Huntingford et al., 2013a). This suggests that in future research, at least for some variables, additional patterns might be added that capture such variability changes, and including any inter-annual variability and adjustment for different warming levels.

In IMOGEN, global temperature changes due to atmospheric gas composition that adjusts radiative forcing is achieved through a small number of parameters in a global energy balance model. However aerosols in particular cause problems for this, as they are not well-mixed, unlike greenhouse gases. Instead, they show strong spatial variation and thus make strong regional variation in radiative forcing. Shiogama et al. (2010) showed that pattern-scaling is less reliable in the case of precipitation

than for temperature, in part because precipitation is more sensitive to aerosol forcing. A potential improvement in the presented method in this regard is to use additional spatial masks for aerosol-affected regions. Other limitations to linear scaling have been identified (Good et al., 2015). For example, local climatic feedbacks are not constant in time, and different components of the climate system respond on different timescales (Chadwick & Good, 2013). This implies that should

IMOGEN be used to test significantly different land surface parameterisations, projected local and regional meteorological changes might no longer be compatible.

Nevertheless, as long as used aware of its limitations, IMOGEN does offer a simple, available and computational fast way to emulate GCMs. This can be operated to estimate surface meteorological changes for different future atmospheric greenhouse pathways. It can also be operated to undertake intermediate analysis with new land surface process descriptions, before their

operation in full-complexity GCMs. This paper takes the further step of adding to its capability the scanning across of a large set of GCMs that it now emulates. Ultimately, the CMIP5 ensemble, which formed the basis of the recent 5th IPCC report (IPCC, 2013) using diagnostics available at that time, has much potential to improve the performance of the described pattern-scaling framework. Aside from the fact that the models themselves have improved, more scenarios are available, allowing better assessment of forcings other than $CO_2$. With preparations now starting for the 6th IPCC report, and new simulations

being made for that, it is timely to consolidate, and calibrate a new set of patterns for the CMIP5 family of GCMs, building on the analysis presented in this paper.

**Data and Code Availability**

The IMOGEN version 2.0 patterns and EBM parameters, along with documentation, are available for full download (under "IMOGEN") from the UK Environmental Information Data Centre (EIDC; http://eidc.ceh.ac.uk). The IMOGEN framework

(Huntingford et al. 2010) have become a component of the JULES land surface initiative (Best et al., 2011; Clark et al., 2011), and it is available via that route (jules.jchmr.org). For the most up-to-date IMOGEN code, please contact the corresponding author.

**Author Contributions**

PZ performed the fitting of the patterns and EBM parameters against the CMIP3 database. CH developed the overall IMOGEN

model framework. LMM advised on impacts applications and NS aided with context placing of the analysis in terms of other GCM emulation systems. All authors contributed towards writing the manuscript.

## Acknowledgements

We acknowledge the modelling groups, the Program for Climate Model Diagnosis and Intercomparison, and the World Climate Research Programme (WCRP) Working Group on Coupled Modelling for their roles in making available the WCRP Coupled Model Intercomparison Project 3 multi-model dataset. This work was financially supported by the Natural Environment Research Council (UK), pilot project Environmental Virtual Observatory (NE/I002200/1), and by the National Science Centre (Poland), grant SONATA 2014/13/D/ST10/00022. CH was also supported by CEH National Capability funds. We thank Seiji Yukimoto of the Climate Research Department, Meteorological Research Institute, Japan, for assistance in processing MRI-CGCM2.3.2 data.

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

**Table 1**. **Availability of WCRP CMIP3 data for the climate patterns study, and characteristics of models' depiction of land and water areas (variable sftlf).** Land/water transition is either continuous (*cont.*) or abrupt (binary mask - *bin.*). The first characteristic in column "Land mask" pertains to coastlines, whereas the second – to inland waters. The codes in column "Data gaps and issues" are: M – missing, E – 4D variable (surface values need to be extrapolated), G – data gaps, R – some data needed resizing.

| | GCM name | Origin | Land mask | Resolution | Data gaps and issues |
|---|---|---|---|---|---|
| 1. | BCCR-BCM2.0 | Norway | cont./cont. | 64x128 | E (HUR) |
| 2. | CGCM3.1(T47) | Canada | bin./bin. | 48x96 | E (HUR) |
| 3. | CNRM-CM3 | France | bin./bin. | 64x128 | E (HUR) |
| 4. | CSIRO-Mk3.0 | Australia | bin./bin. | 96x192 | E (HUR) M (UAS, VAS, PS) |
| 5. | CSIRO-Mk3.5 | Australia | bin./bin. | 96x192 | E (HUR) |
| 6. | GFDL-CM2.0 | USA | cont./bin. | 90x144 | E (HUR) |
| 7. | GFDL-CM2.1 | USA | cont./bin. | 90x144 | E (HUR) |
| 8. | GISS-EH | USA | bin./cont. | 46x72 | E (HUR) |
| 9. | GISS-ER | USA | bin./cont. | 46x72 | E (HUR), M (RLDS) |
| 10. | FGOALS-g1.0 | China | cont./bin. | 60x128 | E (HUR) |
| 11. | INGV-SXG | Italy | bin./bin. | 160x320 | E (HUR, UAS, VAS), M (RLDS, RSDS) |
| 12. | INM-CM3.0 | Russia | bin./bin. | 45x72 | E (HUR) |
| 13. | IPSL-CM4 | France | cont./bin. | 72x96 | E (HUR) |
| 14. | MIROC3.2(hires) | Japan | cont./cont. | 160x320 | E (HUR) |
| 15. | MIROC3.2(medres) | Japan | cont./bin. | 64x128 | E (HUR) |
| 16. | ECHO-G | Germany-Korea | bin/bin. | 48x96 | M (HUR) |
| 17. | ECHAM5/MPI-OM | Germany | cont./bin. | 96x192 | E (HUR) |
| 18. | MRI-CGCM2.3.2 | Japan | cont./cont. | 64x128 | E (HUR) G (PRSN) |
| 19. | CCSM3 | USA | cont./bin. | 128x256 | E (HUR, UAS, VAS) |
| 20. | PCM | USA | cont./bin. | 64x128 | E (HUR, UAS, VAS) |
| 21. | UKMO-HadCM3 | UK | bin/bin. | 73x96 | E (HUR) |
| 22. | UKMO-HadGEM1 | UK | cont./bin. | 145x192 | E (HUR), R (UAS,VAS) |

**Table 2. Calibration parameters of the simple IMOGEN Energy Balance Model, for each considered Global Circulation Model.** The first column presents which runs (experiments) were used to derive the parameters, listed in the following columns: (i) an ocean effective thermal diffusivity which determines the uptake of energy, $\kappa$ (Wm$^{-1}$K$^{-1}$), (ii) a constant ratio of mean land and ocean surface (SST) rate of warming, $\nu$, (iii-iv) climate sensitivity over land $\lambda_l$ and ocean $\lambda_o$ (W m$^{-2}$ K$^{-1}$), and (v) $f$, which is a land fraction, including Antarctica, $f$. The last column presents GCM-specific ratios of warming aver all land per degree of global warming.

| | GCM | Calibration basis | Pattern basis | λl | λo | κ | ν | f | ΔT₁/°K |
|---|---|---|---|---|---|---|---|---|---|
| 1. | BCCR-BCM2.0 | pictrl, 1% to2x | SRES A2 | 2.00 | 2.30 | 350 | 1.40 | 0.28 | 1.26 |
| 2. | CGCM3.1(T47) | pictrl, 1% to4x | SRES A2 | 1.50 | 1.30 | 270 | 1.50 | 0.31 | 1.30 |
| 3. | CNRM-CM3 | pictrl, 1% to4x | SRES A2 | 1.65 | 1.58 | 500 | 1.46 | 0.28 | 1.29 |
| 4. | CSIRO-Mk3.0 | pictrl, 1% to2x | SRES A2 | 1.20 | 1.25 | 2800 | 1.69 | 0.29 | 1.41 |
| 5. | CSIRO-Mk3.5 | pictrl, 1% to2x | SRES A2 | 1.35 | 0.80 | 1300 | 1.58 | 0.29 | 1.35 |
| 6. | GFDL-CM2.0 | pictrl, 1% to4x | SRES A2 | 1.15 | 1.70 | 510 | 1.53 | 0.30 | 1.32 |
| 7. | GFDL-CM2.1 | pictrl, 1% to4x | SRES A2 | 1.15 | 2.05 | 460 | 1.58 | 0.30 | 1.35 |
| 8. | GISS-EH | pictrl, 1% to2x | SRES A1B | 1.30 | 1.65 | 520 | 1.48 | 0.29 | 1.30 |
| 9. | GISS-ER | pictrl, 1% to4x | SRES A2 | 1.05 | 1.40 | 1200 | 1.61 | 0.29 | 1.37 |
| 10. | FGOALS-g1.0 | pictrl, 1% to2x | SRES A1B | 1.80 | 2.80 | 11000 | 1.47 | 0.30 | 1.46 |
| 11. | INGV-SXG | pictrl, 1% to4x | SRES A2 | 0.70 | 1.90 | 320 | 1.65 | 0.28 | 1.39 |
| 12. | INM-CM3.0 | pictrl, 1% to4x | SRES A2 | 1.35 | 1.70 | 500 | 1.50 | 0.30 | 1.30 |
| 13. | IPSL-CM4 | pictrl, 1% to4x | SRES A2 | 1.00 | 1.10 | 700 | 1.57 | 0.30 | 1.34 |
| 14. | MIROC3.2(hires) | pictrl, 1% to2x | SRES A1B | 1.00 | 0.70 | 510 | 1.38 | 0.29 | 1.24 |
| 15. | MIROC3.2(medres) | pictrl, 1% to4x | SRES A2 | 0.83 | 1.00 | 720 | 1.57 | 0.29 | 1.35 |
| 16. | ECHO-G | pdctrl, 1% to4x | SRES A2 | 1.05 | 1.80 | 50 | 1.76 | 0.29 | 1.45 |
| 17. | ECHAM5/MPI-OM | pictrl, 1% to4x | SRES A2 | 0.86 | 0.95 | 500 | 1.60 | 0.29 | 1.36 |
| 18. | MRI-CGCM2.3.2 | pdctrl, 1% to4x | SRES A2 | 1.68 | 1.25 | 380 | 1.38 | 0.30 | 1.22 |
| 19. | CCSM3 | pdctrl, 1% to4x | SRES A2 | 1.10 | 1.70 | 1200 | 1.47 | 0.29 | 1.29 |
| 20. | PCM | pdctrl, 1% to4x | SRES A2 | 1.95 | 2.30 | 720 | 1.43 | 0.29 | 1.27 |
| 21. | UKMO-HadCM3 | pictrl, 1% to2x | SRES A2 | 0.40 | 1.85 | 270 | 1.78 | 0.29 | 1.45 |
| 22. | UKMO-HadGEM1 | pictrl, 1% to4x | SRES A2 | 0.92 | 1.46 | 480 | 1.60 | 0.29 | 1.36 |
| | **All** | | | 1.23±0.40 | 1.57±0.51 | 1148±2220 | 1.55±0.11 | 0.29±0.01 | 1.34±0.06 |

**Table 3. Summary of magnitude, range and robustness of climate change patterns across variables and GCMs**. In each table cell, the first value is the mean, followed by the Percentage Variance Explained statistic (PVE, in brackets, underneath). The values in italics are across-ensemble averages used to fill in data gaps (missing variables for some GCMs). In these cases, the PVE statistic was not calculated. Square brackets in the last column denote average PVE (across-variables) calculated for these incomplete sets of variables.. The pre-last row (All GCMs) contains across-ensemble means of the above statistics, plus associated standard deviations. The last row presents an additional diagnostic of the pattern fitting performance - percentage of negative PVE results, which indicate that the one-parameter regression line is a worse fit than the multi-decadal mean.

| | GCM name | TAS | HUR | UAS+VAS | RLDS | RSDS | PR | PRSN | PS | ALL |
|---|---|---|---|---|---|---|---|---|---|---|
| **1.** | BCCR-BCM2.0 | 1.0463 (83.04) | -0.0201 (13.29) | 0.0089 (2.66) | 5.6754 (76.51) | -0.6185 (9.73) | 0.0300 (8.16) | -0.0025 (13.14) | 0.2324 (30.70) | (29.65) |
| **2.** | CGCM3.1(T47) | 1.0561 (85.79) | 0.0974 (19.80) | 0.0197 (8.30) | 6.0605 (87.28) | -0.4722 (18.82) | 0.0448 (18.71) | -0.0015 (16.66) | 0.0730 (21.33) | (34.59) |
| **3.** | CNRM-CM3 | 1.0371 (86.64) | -0.0111 (24.84) | 0.0098 (8.73) | 5.7499 (81.41) | -0.4978 (19.65) | 0.0255 (17.05) | 0.0000 (17.35) | 0.1838 (38.48) | (36.77) |
| **4.** | CSIRO-Mk3.0 | 1.0740 (82.73) | -0.2354 (13.99) | *0.0108* (-) | 5.9023 (84.39) | -0.6072 (24.46) | 0.0165 (9.38) | -0.0032 (25.70) | *0.0947* (-) | [40.11] |
| **5.** | CSIRO-Mk3.5 | 1.0270 (87.73) | -0.2214 (17.04) | 0.0059 (6.93) | 6.2917 (87.92) | -0.3596 (25.23) | 0.0044 (12.42) | -0.0035 (17.22) | 0.0138 (21.94) | (34.56) |
| **6.** | GFDL-CM2.0 | 1.0731 (83.58) | -0.2653 (14.77) | 0.0071 (8.25) | 6.0471 (83.69) | -1.8451 (34.58) | 0.0042 (16.89) | -0.0041 (21.49) | 0.0657 (24.35) | (35.95) |
| **7.** | GFDL-CM2.1 | 1.0794 (79.71) | -0.2310 (17.35) | 0.0113 (9.59) | 6.2023 (82.62) | -1.7225 (33.21) | -0.0017 (17.93) | -0.0067 (18.66) | 0.2057 (29.21) | (36.03) |
| **8.** | GISS-EH | 1.0543 (75.71) | 0.0080 (10.48) | 0.0103 (4.25) | 6.6734 (77.09) | -1.4802 (1.95) | 0.0415 (11.28) | -0.0016 (8.04) | 0.0264 (12.18) | (25.12) |
| **9.** | GISS-ER | 1.0365 (80.07) | -0.1572 (17.25) | 0.0099 (7.08) | *6.1883* (-) | -0.9399 (11.05) | 0.0384 (14.76) | -0.0018 (10.16) | 0.0251 (18.31) | [22.67] |
| **10.** | FGOALS-g1.0 | 1.1180 (83.05) | -0.0980 (4.11) | -0.0031 (1.36) | 6.3137 (82.71) | -0.9748 (9.67) | 0.0205 (6.70) | -0.0010 (15.52) | 0.0836 (13.62) | (27.09) |
| **11.** | INGV-SXG | 1.0863 (88.01) | -0.0209 (22.96) | 0.0128 (5.20) | *6.2418* (-) | *-0.7667* (-) | 0.0184 (10.68) | -0.0067 (19.10) | 0.0217 (18.58) | (35.98) |
| **12.** | INM-CM3.0 | 1.0604 (83.51) | -0.0391 (15.44) | 0.0130 (4.88) | 5.7124 (78.17) | -0.2908 (23.54) | 0.0200 (14.32) | -0.0071 (12.97) | 0.0124 (16.43) | (31.16) |
| **13.** | IPSL-CM4 | 1.1043 (90.18) | -0.6717 (27.46) | 0.0059 (9.92) | 6.1517 (85.26) | 0.0932 (27.85) | 0.0166 (15.72) | -0.0089 (23.73) | 0.1216 (21.31) | (37.68) |
| **14.** | MIROC3.2(hires) | 1.0801 (94.19) | -0.1528 (16.89) | 0.0031 (8.45) | 6.2088 (92.28) | -1.0731 (36.21) | 0.0271 (20.90) | -0.0052 (25.45) | 0.2352 (37.87) | (41.53) |
| **15.** | MIROC3.2(medres) | 1.1281 (91.16) | -0.2819 (15.56) | 0.0062 (9.38) | 6.4687 (88.97) | -1.7130 (38.30) | 0.0272 (22.37) | -0.0032 (23.53) | 0.1417 (30.26) | (39.94) |
| **16.** | ECHO-G | 1.1383 (88.41) | *-0.1256* (-) | 0.0165 (9.54) | 6.5682 (86.84) | -1.4149 (20.41) | 0.0574 (18.77) | -0.0046 (21.54) | -0.0648 (19.65) | [37.88] |
| **17.** | ECHAM5/MPI-OM | 1.0726 (89.81) | -0.2200 (18.66) | 0.0173 (8.35) | 6.1048 (89.35) | -0.3009 (18.78) | 0.0294 (14.08) | -0.0052 (17.45) | 0.0714 (22.61) | (34.89) |
| **18.** | MRI-CGCM2.3.2 | 1.0641 (84.62) | 0.6378 (13.21) | -0.0048 (3.25) | 6.5956 (82.92) | -1.2903 (10.43) | 0.0387 (7.60) | *-0.0063* (-) | 0.0979 (17.37) | [31.34] |
| **19.** | CCSM3 | 1.1182 (87.92) | -0.0307 (15.09) | 0.0091 (6.75) | 6.8102 (89.67) | -1.2080 (24.48) | 0.0477 (21.47) | -0.0071 (20.45) | 0.1629 (25.02) | (36.36) |
| **20.** | PCM | 1.1059 (78.64) | 0.0704 (9.65) | 0.0346 (1.13) | 6.2678 (75.20) | -0.9678 (8.34) | 0.0573 (12.52) | 0.0009 (13.56) | 0.1509 (17.32) | (27.04) |
| **21.** | UKMO-HadCM3 | 1.0572 (86.57) | -0.7051 (29.40) | 0.0137 (9.85) | 5.6891 (84.81) | 0.2089 (27.15) | 0.0096 (16.36) | -0.0036 (15.88) | -0.0542 (23.05) | (36.63) |

| | | | | | | | | | |
|---|---|---|---|---|---|---|---|---|---|
| **22.** UKMO-HadGEM1 | 1.1222 (88.60) | -0.0900 (18.05) | 0.0190 (15.49) | 6.2180 (89.33) | -1.9194 (37.04) | 0.0076 (21.45) | -0.0039 (22.27) | 0.1814 (32.21) | (40.55) |
| **All GCMs** | 1.079 ±0.032 (85.44 ±4.37) | -0.1256 ±0.2579 (16.92 ±5.71) | 0.0108 ±0.0080 (7.11 ±3.32) | 6.188 ±0.400 (84.74 ±4.97) | -0.9164 ±0.6000 (23.29 ±10.23) | 0.0264 ±0.0165 (14.98 ±4.61) | -0.0040 ±0.0025 (17.96 ±4.67) | 0.095 ±0.084 (23.33 ±6.99) | (34.25 ±5.21) |
| *% neg. PVE* | 0.07 % | 3.87 % | 4.87 % | 0.07 % | 3.64 % | 3.99 % | 1.50 % | 3.37 % | 2.67 % |

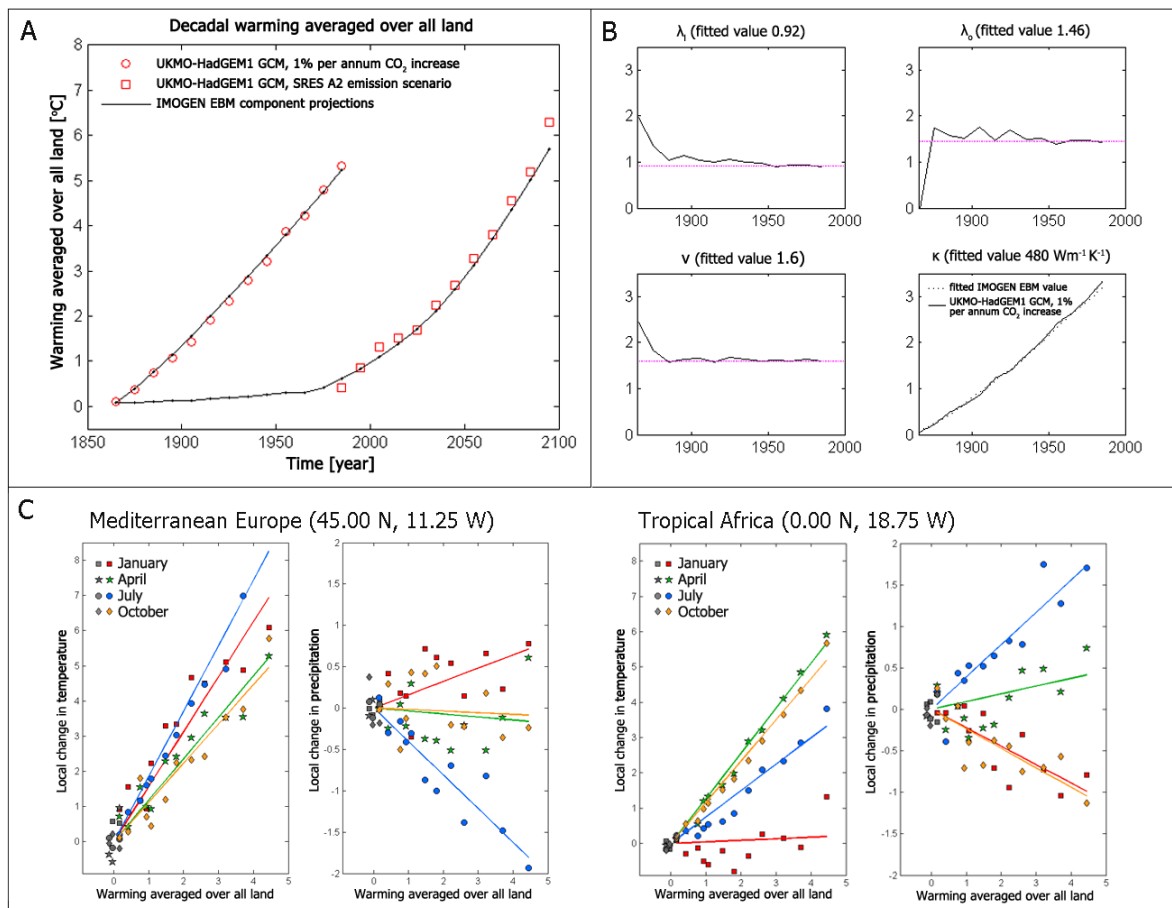

**Figure 1: Illustration of the process leading to parameterization of the IMOGEN Energy Balance Model and the scaled climate patterns (together forming the "pattern scaling set"), based on the example of the UKMO-HadGEM1 GCM. A**: emulation of a warming pathway across time. The 1% to quadrupling atmospheric $CO_2$ run was used for calibration of the energy balance model while the SRES A2 scenario run was used to validate the results. **B**: Fitting of the individual EBM parameters, underlining the match presented in A. Climate sensitivities over land $\lambda_l$ and ocean $\lambda_o$, as well as the ratio of land to ocean warming rate, $\nu$, (pink lines) are derived directly from GCM run data (black curves, 1% run). The fourth parameter, an ocean effective thermal diffusivity, $\kappa$, determines modelled oceanic temperature profile. The $\kappa$ value is selected based upon comparing calculated values of top-of-profile temperature against global mean SST changes projected by UKMO-HadGEM1 1% run (shown). **C**: Example local fitting of patterns of temperature and precipitation, found as regression coefficients (coloured straight lines) against calculated changes in mean temperature over land from UKMO-HadGEM1. Two representative grid-boxes in Mediterranean Europe and Tropical Africa are shown. Coloured symbols are decadal mean monthly values from the UKMO-HadGEM1 SRES A2 run, whilst the grey markers represent data from the 20C3M simulation, which were used to normalize to temperature and precipitation change, and are also corresponding to CRU normals (years 1961–90). Regression "pattern" fit is forced through [0,0] point, as in diagrams.

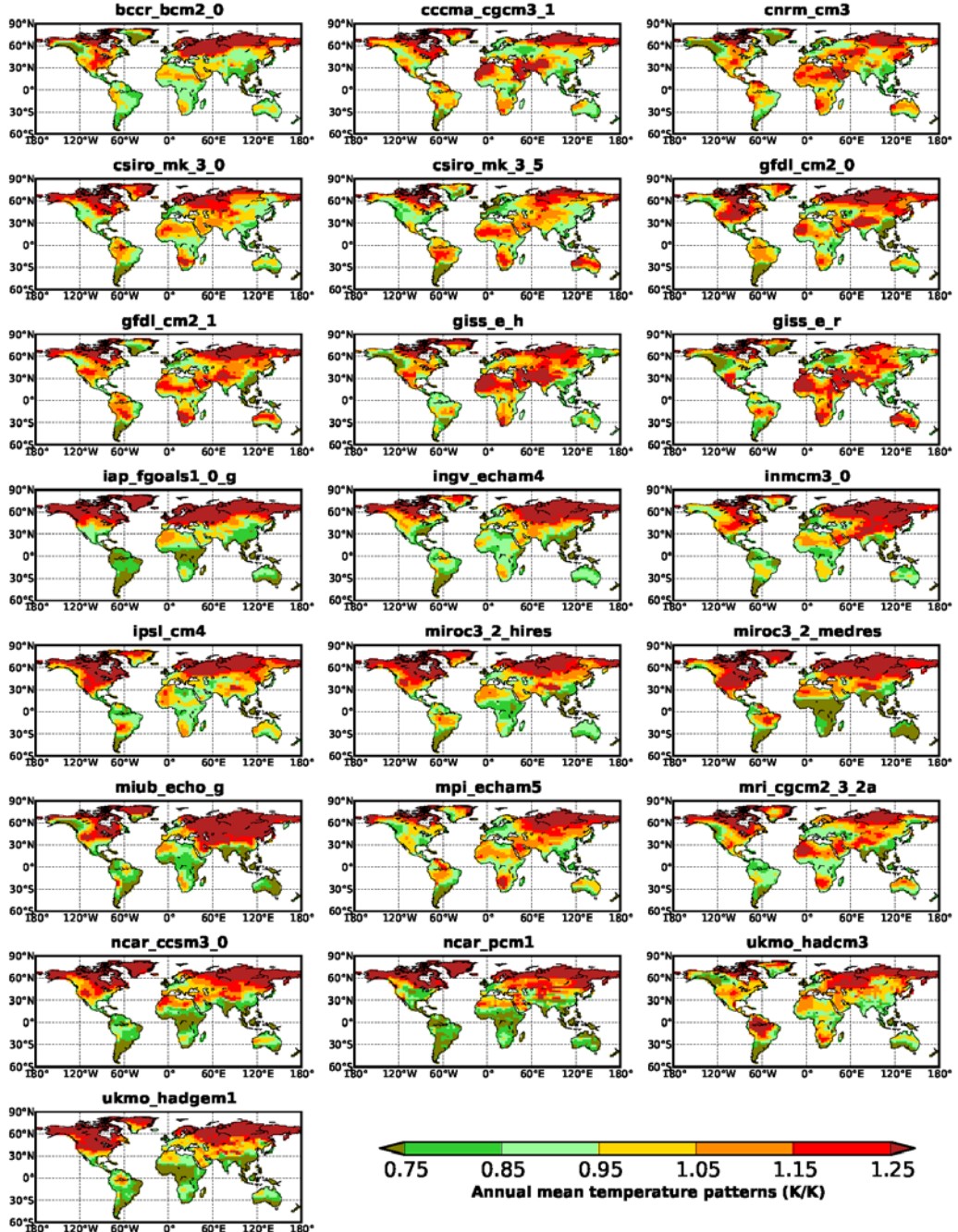

**Figure 2: Annual means of the monthly patterns of local temperature change per degree warming over all land (K K⁻¹).** Data presented for the 22 GCMs considered in this study.

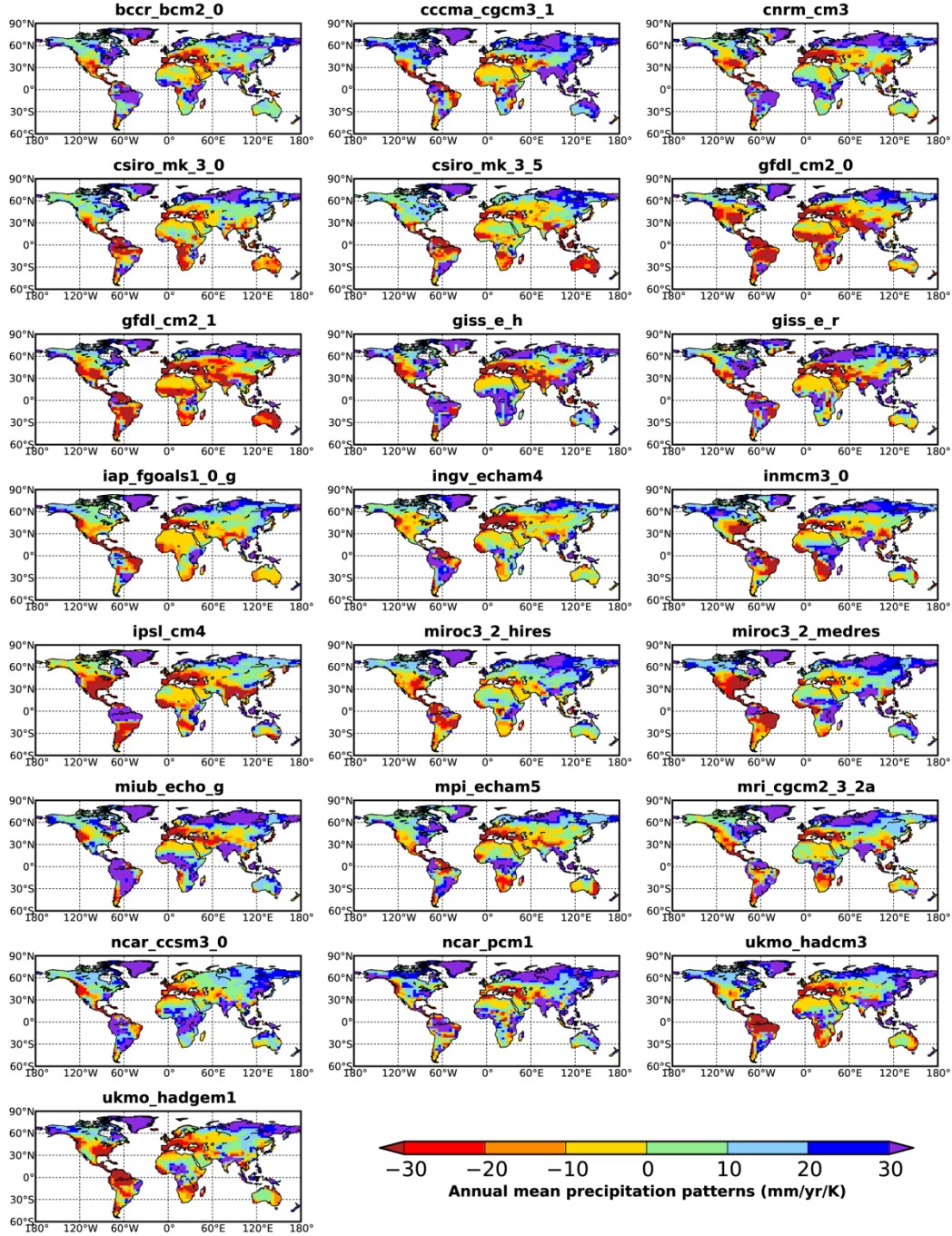

**Figure 3: Annual means of the monthly patterns of local precipitation change per degree warming over all land (mm yr⁻¹ K⁻¹).** Data presented for the 22 GCMs considered in this study.

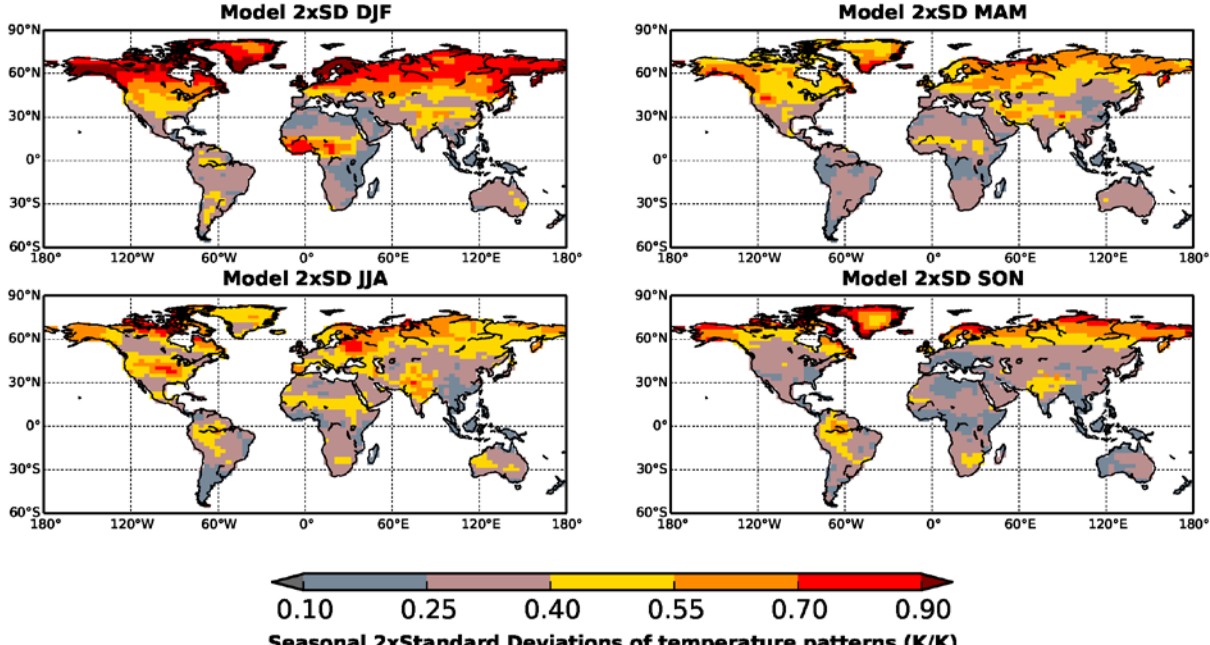

**Figure 4: Seasonal means and variation (2*SD) of the monthly patterns of local temperature change per degree warming over all land (K K$^{-1}$), across 22 GCMs.** DJF is December, January and February, MAM is March, April and May, JJA is June, July and August and SON is September, October and November.

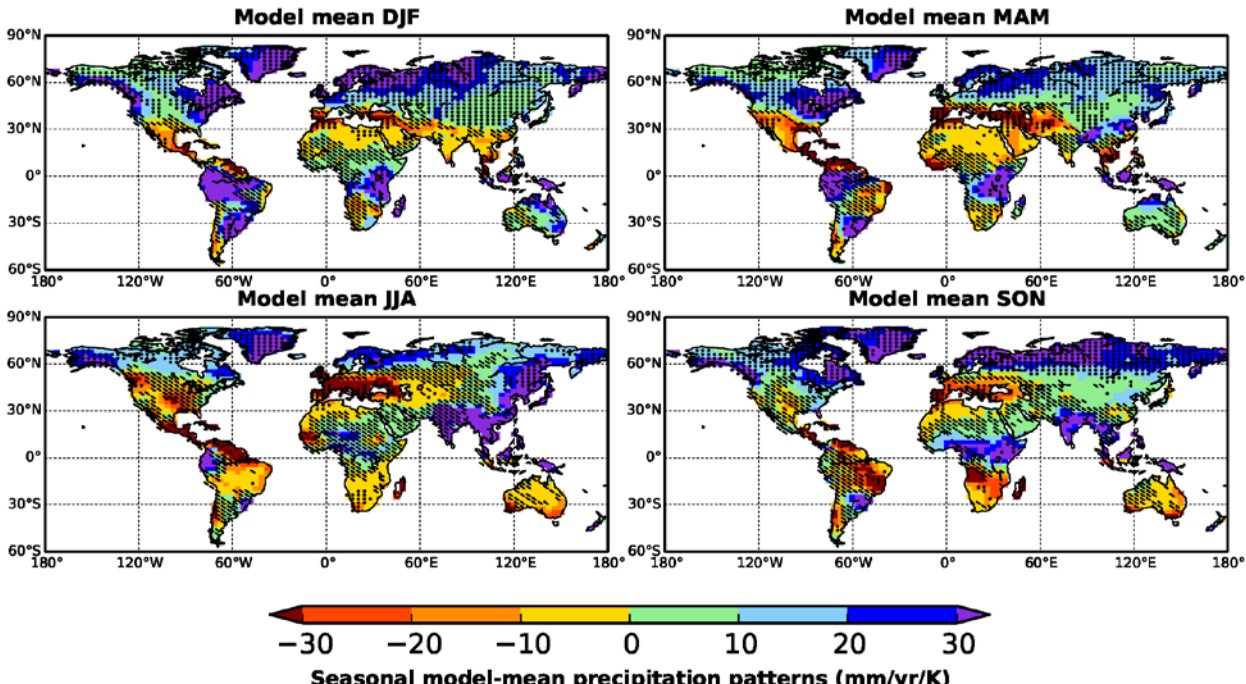

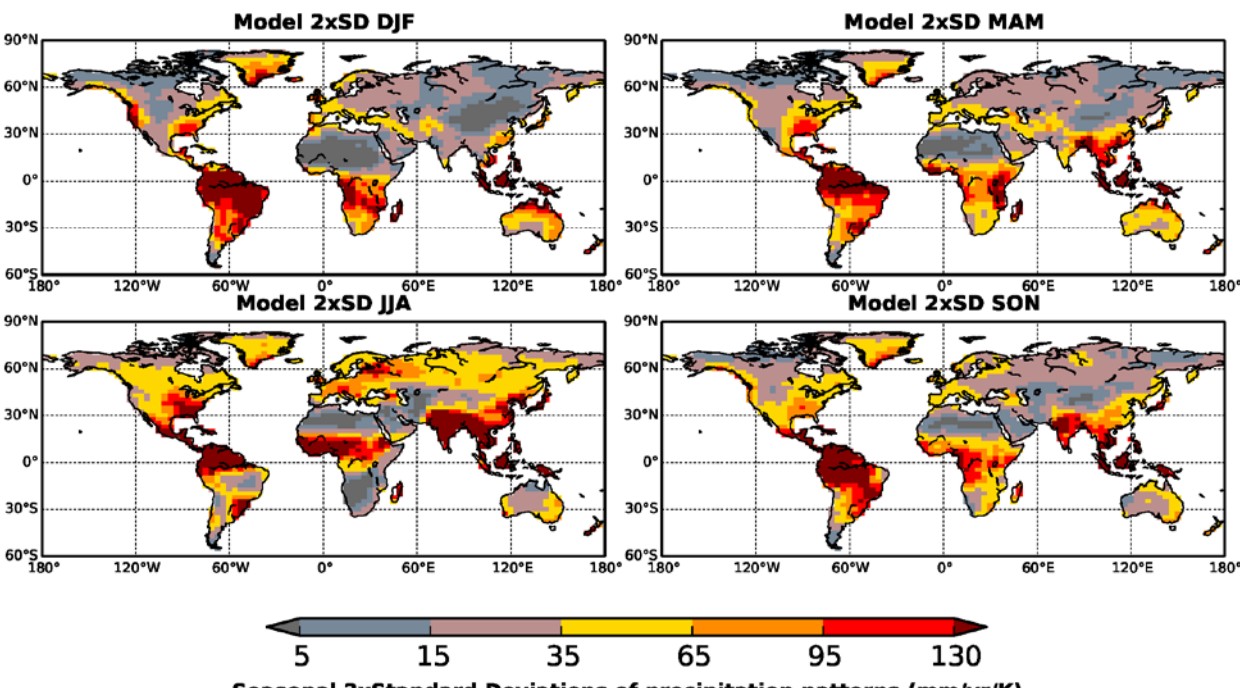

**Figure 5: Seasonal means and variation (2*SD) in the monthly patterns of local precipitation change per degree warming over all land (mm yr-1 K-1), across 22 GCMs.** In regions marked with stippling more than 66% of the models agree in the sign of the change. DJF

is December, January and February, MAM is March, April and May, JJA is June, July and August and SON is September, October and November.

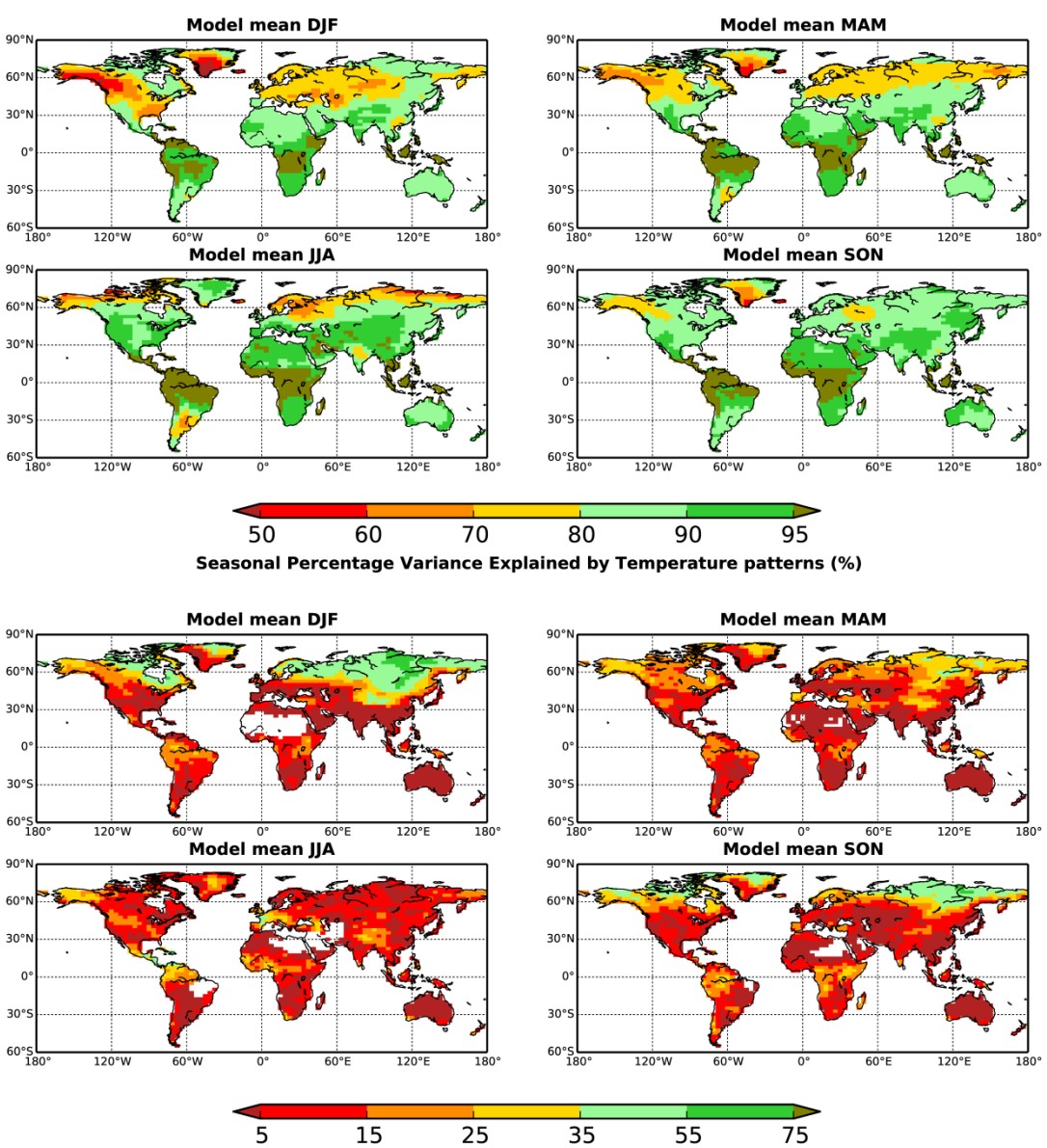

5 **Figure 6: Seasonal percentage of variance explained of the monthly patterns of local temperature and precipitation change per degree warming, across 22 GCMs.** DJF is December, January and February, MAM is March, April and May, JJA is June, July and August and SON is September, October and November.

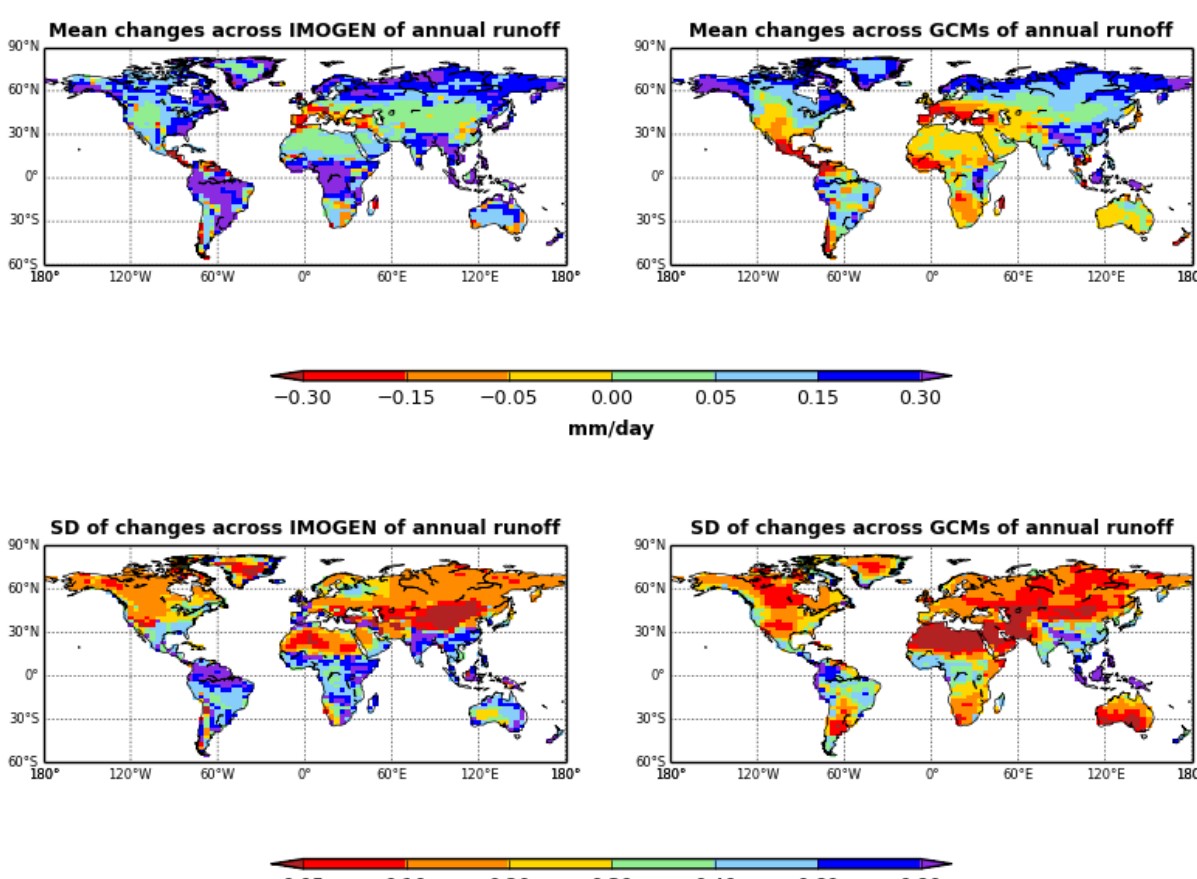

**Figure 7. Comparison of IMOGEN and GCM estimates of annual mean runoff changes** This is for 20 years centred 2090 minus 20 years centred 1900, and for emissions scenario SRESA1B. Top left panel is the mean changes in runoff across GCMs emulated in the IMOGEN system, all forcing the JULES land surface model. Top right panel is the mean changes in runoff as taken directly from the GCMs themselves. In the bottom left panel, at each gridbox, presented is the SD of changes in runoff for IMOGEN, again across GCMs emulated. Bottom right panel is SD of changes in runoff taken directly from the GCMs.