# Peer review of "Climate pattern scaling set for an ensemble of 22 GCMs – adding uncertainty to the IMOGEN impacts system"

_Geoscientific Model Development, 2016_

## Short Comment (SC1) · 30 Nov 2016

Dear authors,

in my role as Executive editor of GMD, I would like to bring to your attention our Editorial version 1.1:

http://www.geosci-model-dev.net/8/3487/2015/gmd-8-3487-2015.html

This highlights some requirements of papers published in GMD, which is also available on the GMD website in the 'Manuscript Types' section:

http://www.geoscientific-model-development.net/submission/manuscript_types.html

[Figure]

In particular, please note that for your paper, the following requirements have not been met in the Discussions paper:

- "The main paper must give the model name and version number (or other unique identifier) in the title."

- "If the model development relates to a single model then the model name and the version number must be included in the title of the paper. If the main intention of an article is to make a general (i.e. model independent) statement about the usefulness of a new development, but the usefulness is shown with the help of one specific model, the model name and version number must be stated in the title. The title could have a form such as, "Title outlining amazing generic advance: a case study with Model XXX (version Y)"."

- "All papers must include a section, at the end of the paper, entitled 'Code availability'. Here, either instructions for obtaining the code, or the reasons why the code is not available should be clearly stated. It is preferred for the code to be uploaded as a supplement or to be made available at a data repository with an associated DOI (digital object identifier) for the exact model version described in the paper. Alternatively, for established models, there may be an existing means of accessing the code through a particular system. In this case, there must exist a means of permanently accessing the precise model version described in the paper. In some cases, authors may prefer to put models on their own website, or to act as a point of contact for obtaining the code. Given the impermanence of websites and email addresses, this is not encouraged, and authors should consider improving the availability with a more permanent arrangement. After the paper is accepted the model archive should be updated to include a link to the GMD paper."

If I understood your article correctly, the main model you developed is IMOGEN, therefore a version number of IMOGEN should be given in the title and a code availability

section should inform the reader about if and how to gain access to IMOGEN.

Yours,

Astrid Kerkweg

––––––––––––––––––––––––––––

---

## Referee Comment (RC1) · S. Emori (Referee) · 15 Dec 2016

The paper describes a system for emulating CMIP3 climate models in terms of land surface variables at low computational cost, including its design, performance, applications and limitations. The manuscript is clearly written. I understand the importance of the attempt and usefulness of the tool, although honestly I felt that CMIP3 models are relatively dated and the authors might have been able to adopt CMIP5 data sooner.

I have a couple of overall comments, which are rather my impression and not something really critical.

1. In my view, a climate emulator like the one presented here based on an energy balance model and pattern scaling is particularly powerful when it is applied for exploring a wide range of different scenarios (e.g., intermediate levels between RCPs). If the authors agree with this, it should be mentioned with more emphasis.

2. The applications described are mostly focused on ecosystem impacts. I reckon however that the tool has a potential to be applied to a wider range of impact studies, including water resource, agriculture, health and so on. It could also be emphasized.

Minor comments:

1. P.2, L.2, "Global Climate Models (GCMs, also called Earth System Models, ESMs)": In my understanding, a GCM is called an ESM especially when (or only if) it incorporates some biogeochemical components. Simply paraphrasing them sounds uncomfortable to me.

2. P.3, L.1, "Intergovernmental Panel for Climate Change": "on" instead of "for".

3. P.4, L.33, "50 km (e.g. MIROC3.2hires model,": The atmospheric resolution of MIROC3.2-hires is T106, which is approximately 100 km in mid-latitudes instead of 50 km.

4. P.5, L.14, "climate regime c": Not clear what it means.

5. P.5, L.22, "one regression co-efficient, rather than two." Not entirely clear to me. Do you mean only slope rather than slope and intercept (or, equivalently, intercept is always zero)?

6. P.5, L.33, "(ii) a constant ratio of mean land and ocean surface (SST) rate of warming, $\nu$, (iii-iv) climate sensitivity over land $\lambda l$ and ocean $\lambda o$ (W m-2 K-1)": Instinctively, it sounds over-specification to me, as it looks like v could be calculated by $\lambda l/\lambda o$ (at least approximately). It might be my silly misunderstanding, but a bit of further explanation might be helpful to other readers as well.

7. P.6, L.13, Eq (2): "(is, ms, g)" in the r.h.s might have to be "(gs, ms, i)".

8. P.16, L.22: "Shiogama, H., Shiogama, H.,": duplicated. Please delete one.

---

## Referee Comment (RC2) · Anonymous Referee #2 · 19 Dec 2016

GMD-2016-221 Zelazowski et al., Climate Pattern Scaling Set for an Ensemble of 22 GCMs

As it stands, this paper is a simple implementation of the idea of pattern scaling and EMB calibration to a multi-model ensemble. I do not think any of this is new per se, and the choice of presenting this work based on a by now obsolete CMIP ensemble, CMIP3, does not help making the case for publication as is. That said, I think the value of the work resides in the provision of a multivariate set of fields/patterns, for the use in impact work, effectively exemplified by the motivating application through the IMOGEN system. This is in my opinion the real contribution of this work, but the focus of the paper is not adequately trained on this aspect.

[Figure]

I would recommend major revisions, but I would hope that my request would not be a show stopper: what I would like to see is the application section expanded, not because I want to see the results of the impact analysis for their own value, but because I think we need to see how the pattern scaling performs in the application context, compared to results obtained from using the actual GCM output (by the way, I do think the use of ESM here is not appropriate, I don't think any of the CMIP3 models was an ESM in the sense of including a representation of the carbon cycle). My main concern is to be able to assess how the differential performance of the pattern scaling approach across variables and across models impacts the results of a multi-model impact assessment. I do not think the application section at this time addresses that. A particular concern is how the performance on individual variables translates into a performance across variables, i.e., in their joint behavior, for different models' output. In fact, in this regard, even the section about "Explanatory power of linear approximation" needs a better description: What is the meaning of the sentence (and I summarize) "Overall, climate patterns explain one third of regional climate change". How is the joint variability/covariability of the variables evaluated? Is the covariance patterns among all variables taken into account? I would like to see a more rigorous and formal definition of how the variance of the joint set of variables is represented by the emulation.

If the authors are willing to show how the use of the pattern scaling solution compares to the use of the original output from the multi-model ensemble I think the article will become more informative and valuable to the impact research community, within and beyond IMOGEN users. In this respect I also agree with Dr. Emori that the potential is larger than just the IMOGEN application and it would be good to point that out.

Last, two very minor points: I think throughout the paper the word "assembly" has been erroneously substituted for "ensemble", my guess because of an auto-correct program. The other word, which I think is used instead deliberately but I question, is "meteorology". I think what the pattern scaling approach produces is still "climatology". These are after all ten-year means. The use of a weather generator may then produce

meteorology at the time scale needed by the impact model, but that is an add-on to the method that this paper focuses on.

---

## Author Comment (AC1) · 1 Feb 2017

Please note that some internal numbering from the review has changed slightly as review comments have been incorporated.

Comment 1.1. Dear authors, in my role as Executive editor of GMD, I would like to bring to your attention our Editorial version 1.1: http://www.geosci-model-dev.net/8/3487/2015/gmd-8-3487-2015.html. This highlights some requirements of papers published in GMD, which is also available on the GMD website in the 'Manuscript Types' section: http://www.geoscientific-model development.net/submission/manuscript_types.html

[Figure]

Response 1.1. Thank you for this and other comments. We have familiarised ourselves with the Editorial.

C.1.2. In particular, please note that for your paper, the following requirements have not been met in the Discussions paper: The main paper must give the model name and version number (or other unique identifier) in the title.

R.1.2. We have added the version number, and as IMOGEN vn 2.0. This makes a clear distinction from vn 1.0, which was only calibrated against the United Kingdom Met Office model, HadCM3. This distinction is also noted in the main body of the text.

C.1.3. If the model development relates to a single model then the model name and the version number must be included in the title of the paper. If the main intention of an article is to make a general (i.e. model independent) statement about the usefulness of a new development, but the usefulness is shown with the help of one specific model, the model name and version number must be stated in the title. The title could have a form such as, "Title outlining amazing generic advance: a case study with Model XXX (version Y)".

R.1.3. Please see response to request above. IMOGEN v2.0 corresponds to the new calibration against a set of 22 GCMs. This paper is presenting a new model version.

C.1.4. All papers must include a section, at the end of the paper, entitled 'Code availability'. Here, either instructions for obtaining the code, or the reasons why the code is not available should be clearly stated. It is preferred for the code to be uploaded as a supplement or to be made available at a data repository with an associated DOI (digital object identifier) for the exact model version described in the paper. Alternatively, for established models, there may be an existing means of accessing the code through a particular system. In this case, there must exist a means of permanently accessing the precise model version described in the paper. In some cases, authors may prefer to put models on their own website, or to act as a point of contact for obtaining the code. Given the impermanence of websites and email addresses, this is not encouraged, and authors should con- sider improving the availability with a more permanent arrangement. After the paper is accepted the model archive should be updated to include a link to the GMD paper.

R.1.4. The main basis for the paper is the new driving dataset – the patterns and EBM parameters forcing the IMOGEN model, rather than the model itself. However, these developments in available drivers are sufficiently large as to constitute a new model version - here named v2.0. The original model is written out in full in existing papers, including an earlier Huntingford et al 2010 GMD paper. Unfortunately our original paper version had the link to our archived model drivers named "Data Availability". This is now renamed "Data and Code Availability".

The driving dataset described in this paper was placed on a permanent and well-recognised international archive EIDC, and is now linked appropriately from the manuscript. In addition, we now also cite the 2010 GMD paper regarding the model description, and add: "For current IMOGEN code, please contact the corresponding author" (page 12, from line 26).

C.1.5. If I understood your article correctly, the main model you developed is IMOGEN, therefore a version number of IMOGEN should be given in the title and a code avail-ability section should inform the reader about if and how to gain access to IMOGEN.

R.1.5. Agreed; and implemented as explained above, including within the title "IMO-GEN vn2.0".

---

## Author Comment (AC2) · 1 Feb 2017

Comment 2.1. The paper describes a system for emulating CMIP3 climate models in terms of land surface variables at low computational cost, including its design, performance, applications and limitations. The manuscript is clearly written. I understand the importance of the attempt and usefulness of the tool, although honestly I felt that CMIP3 models are relatively dated and the authors might have been able to adopt CMIP5 data sooner. I have a couple of overall comments, which are rather my impression and not something really critical.

Response 2.1. We thank the referee for this review and a number of useful comments. We accept that a dataset based on the CMIP5 dataset is required. However, as we

outline in the paper, a number of studies are based on the combination of the IMOGEN model and the presented set of patterns. This paper is intended to describe in a greater detail the dataset and how it was derived. We intend to work with an equivalent dataset based on CMIP5 and believe that there will be a benefit that we can refer back to the dataset described here, allowing comparison.

C.2.2. In my view, a climate emulator like the one presented here based on an energy balance model and pattern scaling is particularly powerful when it is applied for exploring a wide range of different scenarios (e.g., intermediate levels between RCPs). If the authors agree with this, it should be mentioned with more emphasis.

R.2.2. Agreed; we have already mentioned this briefly in the abstract, and in the beginning of the Introduction, but now we have expanded the later according to this suggestion (from line 9). In the Introduction, we now write:

"It allows interpolation away from a limited number of available GCM simulations, enabling a time-efficient assessment of surface meteorological changes for alternative non-standard future scenarios of changed GHG concentrations. This can include, for example, new scenarios that fall between the current Representative Concentration Pathways (RCP, Taylor et al., 2012), and potentially to investigate the current focus on targeting pre-defined future temperature thresholds such as two degrees"

C.2.3. The applications described are mostly focused on ecosystem impacts. I reckon however that the tool has a potential to be applied to a wider range of impact studies, including water resource, agriculture, health and so on. It could also be emphasized.

R.2.3. Agreed, and some of these potential applications we hope to pursue, so having IMOGEN v2.0 published will be helpful. Combining this comment with the suggestion from the Referee #2, we have added a new part to Section 4 "Applications" in which we undertake an assessment of IMOGEN performance in terms of ability to project changes to impacts, rather than just the direct climatic changes. We focus on mean annual total runoff, and making a direct comparison to GCM estimates of change. We

also added a new figure (#7). Please see response 3.1 below.'

Minor comments:

C.2.4. P.2, L.2, "Global Climate Models (GCMs, also called Earth System Models, ESMs)": In my understanding, a GCM is called an ESM especially when (or only if) it incorporates some biogeochemical components. Simply paraphrasing them sounds uncomfortable to me.

R.2.4. Agreed. Following this suggestion, and the one from Referee #2 (3.1.), we have dropped the use of "Earth System Model" description in the case of the analysed CMIP3 data.

C.2.5. P.3, L.1, "Intergovernmental Panel for Climate Change": "on" instead of "for".

R.2.5. Corrected.

C.2.6. P.4, L.33, "50 km (e.g. MIROC3.2hires model,": The atmospheric resolution of MIROC3.2-hires is T106, which is approximately 100 km in mid-latitudes instead of 50 km.

R.2.6. Corrected.

C.2.7. P.5, L.14, "climate regime c": Not clear what it means.

R.2.7. Changed to "climate regime in the decade c"

C.2.8. P.5, L.22, "one regression co-efficient, rather than two." Not entirely clear to me. Do you mean only slope rather than slope and intercept (or, equivalently, intercept is always zero)?

R.2.8. Yes, that is what we meant, and this is now stated more clearly in the same place. We now write: "This implies that the regression line starts at the origin of the co-ordinate system, so the intercept equals zero, and there is a fit with just one regression co-efficient, the slope"

C.2.9. P.5, L.33, "(ii) a constant ratio of mean land and ocean surface (SST) rate of warming, $\nu$, (iii-iv) climate sensitivity over land $\lambda$l and ocean $\lambda$o (W m-2 K-1)": Instinctively, it sounds over-specification to me, as it looks like v could be calculated by $\lambda$l/$\lambda$o (at least approximately). It might be my silly misunderstanding, but a bit of further explanation might be helpful to other readers as well.

R.2.9. Whilst we agree that having individual climate sensitivities over land and ocean might initially seem an over-specification, the concern we had is that the ocean component of the global energy balance behaves very differently to the land surface. The ocean is a huge store of thermal energy, currently making the planet lag significantly - by decades - behind the true level of committed warming for current atmospheric GHG concentrations. Relative to the oceans, the land operates so as to have almost negligible thermal capacity, and hence we wanted to capture both effects individually. This also suggests thermal energy flows by atmospheric transport from the land regions to the oceans regions. Initially in Huntingford and Cox (2000), we tested this by modelling an advection term k*(dT_air-dT_ocean), but the fit was poor. Instead we found we could close the equations, capturing GCMs well, with simply a land/atmosphere fixed contrast. This gives an implicit advection, with "k" being a function of time. It is still an area of open research as to why both historical measurements and GCMs all project near-constant (in time) land/atmosphere contrasts.

C.2.10. P.6, L.13, Eq (2): "(is, ms, g)" in the r.h.s might have to be "(gs, ms, i)".

R.2.10. Agreed – thank you very much for spotting this mistake.

C.2.11. P.16, L.22: "Shiogama, H., Shiogama, H.,": duplicated. Please delete one.

R.2.11. Deleted

---

## Author Comment (AC3) · 1 Feb 2017

Comment 3.1. As it stands, this paper is a simple implementation of the idea of pattern scaling and EMB calibration to a multi-model ensemble. I do not think any of this is new per se, and the choice of presenting this work based on a by now obsolete CMIP ensemble, CMIP3, does not help making the case for publication as is. That said, I think the value of the work resides in the provision of a multivariate set of fields/patterns, for the use in impact work, effectively exemplified by the motivating application through the IMOGEN system. This is in my opinion the real contribution of this work, but the focus of the paper is not adequately trained on this aspect. I would recommend major revisions, but I would hope that my request would not be a show stopper: what I would

like to see is the application section expanded, not because I want to see the results of the impact analysis for their own value, but because I think we need to see how the pattern scaling performs in the application context, compared to results obtained from using the actual GCM output (by the way, I do think the use of ESM here is not appropriate, I don't think any of the CMIP3 models was an ESM in the sense of including a representation of the carbon cycle). My main concern is to be able to assess how the differential performance of the pattern scaling approach across variables and across models impacts the results of a multi-model impact assessment. I do not think the application section at this time addresses that.

Response 3.1. We thank the Referee for this constructive criticism. First, on notation and similar to the request of referee one, we have dropped the use of "Earth System Model" description in the case of the analysed CMIP3 data.

Based on this comment, we have expanded significantly Section 4 "Applications" in which we undertake an assessment of IMOGEN performance in terms of ability to project changes to impacts. The focus is placed on mean annual total runoff, and making a direct comparison to GCM estimates of change. We also added a new Figure 7. The new diagram, its caption and associated new text in main body of the paper is repeated below.

Figure 7 (Caption – figure below): Estimates of gridbox mean annual total runoff, RTot (mm/day). These are: top panel, for IMOGEN and year 1860: middle panel, IMOGEN estimates and year 2090 calculations of RTot minus those of year 1860: bottom panel, HadCM3 estimates and mean of year 2080-2099 calculations of RTot minus those of mean of last 20 years of pre-industrial control simulation.

New text: "We additionally undertake an assessment of IMOGEN performance in terms of ability to project changes to impacts, and when compared directly to GCM estimates of change. Many of the components of the land surface component of IMOGEN, i.e. JULES, remain similar to those operated in the HadCM3 GCM. Hence we evaluate an

IMOGEN simulation operated with the HadCM3 patterns, by assessing performance against terrestrial diagnostics directly from the HadCM3 model. For both IMOGEN and HadCM3 simulations, this is with SRESA2 $CO_2$ emissions and estimated non-$CO_2$ radiative forcing also for that scenario, and with the GCM calculations drawn from the CMIP3 database. The variable we select is total runoff, which is the combination of surface and subsurface runoff calculations. This is available from both IMOGEN and HadCM3, and here presented as annual gridbox mean value, RTot (mm day-1).

Runoff provides a challenge for comparison, as it is frequently a relatively small number between two larger fluxes of precipitation and evapotranspiration (transpiration, plus soil evaporation and interception loses) and so sensitive to change in those fluxes. Direct comparison also needs to account for IMOGEN being initialised with a climatology based on the CRU dataset, and temporal dis-aggregation to sub-daily drivers of JULES having not been calibrated against any particular GCM. Nevertheless, to be a useful tool for impacts assessment, then IMOGEN must capture the general features of GCM projections when operated for similar emissions scenarios.

In Figure 7, we compare IMOGEN versus HadCM3 projections of change in RTot. The top panel is modelled year 1860 values, from IMOGEN. The middle panel is the change in RTot, again for IMOGEN, and between years 1860 and 2090. The bottom panel is the change in RTot for HadCM3, comparing the last 20 years of the pre-industrial control simulation against the last 20 years of SRES-A2 forced simulation, which for the latter is 2080-2099. Multi-year averages are derived to remove any inter-annual variability, which as yet, IMOGEN does not represent. Although there are apparent local differences, and recognising the caveats above, then at its most general many dominant geographical features of change in IMOGEN do have similarities to those of HadCM3."

C.3.2. A particular concern is how the performance on individual variables translates into a performance across variables, i.e., in their joint behavior, for different models' output. In fact, in this regard, even the section about "Explanatory power of linear

approximation" needs a better description: What is the meaning of the sentence (and I summarize) "Overall, climate patterns explain one third of regional climate change". How is the joint variability/covariability of the variables evaluated? Is the covariance patterns among all variables taken into account? I would like to see a more rigorous and formal definition of how the variance of the joint set of variables is represented by the emulation.

R.3.2. We agree that the issue of co-variability is important, and are aware of the studies that look in to more complex pattern-scaling models which partly address this issue (e.g. Frieler et al 2012, now cited in this manuscript). We request, that on this one issue, this is beyond the scope of this presentation of the IMOGEN 2.0 model.

However we do acknowledge some poor wording in the paper. First, the section of concern is now titled: "3.3. Performance of linear approximation assumption in "pattern-scaling" for individual variables". Then, in that Section, we now write more clearly, adding "when per-variables results are averaged" as: "Overall (i.e. when per-variable results are averaged, without considering co-variance), climate patterns explain one-third of regional climate change (PVE 34.25±5.21)"

C.3.3. If the authors are willing to show how the use of the pattern scaling solution compares to the use of the original output from the multi-model ensemble I think the article will become more informative and valuable to the impact research community, within and beyond IMOGEN users. In this respect I also agree with Dr. Emori that the potential is larger than just the IMOGEN application and it would be good to point that out.

R.3.3. We agree, and have expanded extensively our Section 4 "Applications", with a focus on change in mean annual total runoff. This is by a direct comparison between GCM estimates of change (for HadCM3), and IMOGEN estimates. Please see our response to query C.3.1. above, including listing of new diagram, caption and additional text within the manuscript.

C.3.4. Last, two very minor points: I think throughout the paper the word "assembly" has been erroneously substituted for "ensemble", my guess because of an auto-correct program.

R.3.4. Corrected – thank you very much for noticing this mistake.

C.3.5. The other word, which I think is used instead deliberately but I question, is "meteorology". I think what the pattern scaling approach produces is still "climatology". These are after all ten-year means. The use of a weather generator may then produce meteorology at the time scale needed by the impact model, but that is an add-on to the method that this paper focuses on.

R.3.5. We agree that "climatology" is more accurate and would like to explain that originally we referred to "meteorology" because of IMOGEN's weather generator. Six instances of "meteorology" have been changed throughout the text to "climatology".

[Figure]

**Fig. 1.** Figure 7: Estimates of gridbox mean annual total runoff, RTot (mm/day). These are: top panel, for IMOGEN and year 1860: middle panel, IMOGEN estimates and year 2090 calculatio

---

## Author Comment (AC4) · 10 Sep 2017

Please find uploaded a cover letter that describes our new calculations and additional figure in response to reviewer request to analyse IMOGEN projections of an impact, and versus direct GCM outputs. Detailed manuscript changes are listed on page 2 and 3.

Please also note the supplement to this comment: https://www.geosci-model-dev-discuss.net/gmd-2016-221/gmd-2016-221-AC4-supplement.pdf

[Figure]

**Supplement:**

[Figure]

**CEH Wallingford**
Wallingford, OXON, OX10 8BB, U.K.
Email: chg@ceh.ac.uk
Tel: +44 (0)7884437138          September 10th 2017

Prof. Wilco Hazeleger
Editor
Geoscientific Model Development

Dear Prof. Wilco,

Thank you for your help on manuscript titled:

**"Climate patterns scaling set for an ensemble of 22 GCMs - adding uncertainty to the IMOGEN impacts system"**

At the last paper version, we had answered both reviewers' comments to their acceptable level, except for the additional request to test the modelling framework for mean changes and spread in a key impact (Referee #2). Almost all of the climate models have available gridbox mean runoff, and so we used that quantity to answer this final query.

We are grateful for being asked to do this, as it does allow a new perspective on the pattern-scaling approach. In general, we find similarities between IMOGEN projections (which although spanning GCMs emulated, have the common land surface model JULES), and those from direct GCM estimates of runoff. However there are sufficient differences for us to conclude that future efforts on pattern-scaling could include direct scaling of variables such as runoff. That is in addition to the meteorological drivers.

As you know of course, there is presently much interest in global warming pathways to stabilisation at 1.5 and 2.0°C. Unfortunately very few GCM simulations are available that target those precise final temperatures. Pattern-scaling provides a mechanism to achieve that, by interpolating from existing GCM simulations. Hence we hope use will be made of this manuscript, and the related parameters and patterns that are available for download, to advise on expect impacts for either final warming level.

I am sorry it has taken us over two months to respond with this final analysis. This is in part because it took time to return to the original GCM simulations, and to extract the extra runoff variable. On behalf of myself and co-authors, I very much hope our revised manuscript can now be considered acceptable for publication in Geoscientific Model Development. Our full response to reviewer is given below. Changes in the manuscript are highlighted in blue.

With kind regards,

Prof. Chris Huntingford

Dear Reviewer 2 – thank you for your additional request below for manuscript **"Climate patterns scaling set for an ensemble of 22 GCMs - adding uncertainty to the IMOGEN impacts system".** This has allowed us to have a new perspective on IMOGEN pattern-scaling capability.

Reviewer #2 writes:
"Please show an application where all the patterns (i.e. all the CMIP3 models) are used, and show that on average the errors between true and pattern-scaled input to the impact models is within the original multi-model variability/uncertainty from the CMIP3 ensemble. That, in my opinion, is the correct way to validate the utility of this implementation. And if I had to bet, given the variance among CMIP3 models, the application will successfully demonstrate its "accuracy" within the CMIP3 universe of projections."

We have chosen to answer this request via the quantity total gridbox-mean runoff (mm/day). This has led to a new Figure 7 and associated text in the paper. These additional paper components are listed below, and please accept these as our responses. We conclude that the land surface model has a strong influence on runoff projection changes, including the variance. We conclude that future pattern-scaling activities could include direct scaling of variables such as runoff, rather than just surface-level meteorological changes.

**New figure and Caption**:

[Figure]

**Figure 7.** Comparison of IMOGEN and GCM estimates of annual mean runoff changes (20 years centred 2090 minus 20 years centred 1900), and for emissions scenario SRESA1B. Top left panel is the mean changes in runoff across GCMs emulated in the IMOGEN system, all forcing the JULES land surface model. Top right panel is the mean changes in runoff as taken directly from the GCMs themselves. In the bottom left panel, at each gridbox, presented is the SD of changes in runoff for IMOGEN, again across GCMs emulated. Bottom right panel is SD of changes in runoff taken directly from the GCMs.

**Text for main paper:**

In Figure 7, for SRESA1B emissions scenario, we consider the ability of IMOGEN to project runoff changes and compare the result to such changes taken directly from the GCMs themselves. Hence this is comparing the IMOGEN simulations that emulate multiple GCMs but with a single land surface model (JULES), versus runoff values directly from the GCMs. The latter therefore contain alternative estimates of climate change, and additionally the responses of different land surface models. During modelled pre-industrial period, and modelled period centred on year 2090, total runoff values are recorded for each GCM (both emulated in IMOGEN, and directly from GCMs). Then in each case the change is calculated. In the top panels, the mean of these changes are shown, whilst the bottom panels are the standard deviations of these change values. Although there are similarities between left and right-hand panels (over northern latitudes, in particular), there are important differences too (notably the drying signal in GCM output for Africa and Australia is not reflected by the IMOGEN framework). For SDs, in some locations there is higher variability for IMOGEN than for the GCMs themselves; however, this pertains mainly to the regions where IMOGEN predicts higher runoff. The latter may be surprising, as considering that GCMs directly introduce another level of uncertainty i.e. inter-land surface model differences. Our finding is suggestive that JULES has a particularly sensitive response of runoff to imposed climatic changes. Looking ahead to new forms for IMOGEN, one possibility to additionally capture the different land surface responses is to pattern-scale directly impact variables of interest such as runoff.

---

## Editor Decision (ED1)

**[Reviewer Request]** It's clear to me that this work is very late in getting published, probably the first author has moved on to other type of work. That does not justify however the treatment of this submission in such evident cursory manner. The new paragraph inserted repeats the first sentence twice. There is nothing in the discussion that considers this new important result. There are references within the manuscript that still refer to AR4 as if AR5 and the corresponding new understanding of precipitation projection variability hadn't not come to fruition in the meantime. I will not mention the fact that we have cmip3 results in here. So my last request is not for additional work but for some due diligence in going through this article one more time, update references and most importantly clearly discuss the results of the exercise in comparison of runoff, and not by saying " we should pattern scale runoff" too. But by discussing the role of impact models as an additional and important source of variation, and clearly state that what IMOGEN supplies is something that aides the treatment of (by now outdated unfortunately) climate model uncertainty only. User beware.

**[General Response]** Dear reviewer – thank you for agreeing to look at our manuscript one further time. We answer in general terms first and then repeat below this the detailed additional text added to the manuscript.

The request to study IMOGEN projections of an impact of interest – we selected total runoff – has led to a more complete study. Comparing IMOGEN estimates against those directly from the GCMs introduces another level of uncertainty (at present, IMOGEN is only coupled to the JULES land surface model). To build the new Figure 7, illustrating implications for runoff of IMOGEN estimates versus direct GCM estimates, was done carefully and thoroughly. I'm sorry typos appeared when carrying things over to the paper itself. I accept myself responsibility for that.

What we didn't realise is the extent to which the reviewer wanted the discussion of the impact of runoff to be discussed through the paper. We are very happy to now correct for that. It is now mentioned explicitly in the Abstract, and has more prominence in the discussion part of the paper. Having presented the IMOGEN framework at various meetings, then it has been a serious suggestion made by many to actually scale impact changes directly against level of global warming. Hence, I would like to retain that as a discussion point if the reviewer is agreeable. The paper has been re-worded, though, so it no longer appears as a "throwaway" comment.

We have updated the manuscript with more recent and relevant references that we missed. A version of the paper has been uploaded with track changes so all changes can be identified. We respond with details in full to the reviewer below. In addition, we have worked through the paper careful to pick up some final points of clarity and other last small typos. These can also be seen in track changes of the new version.

**[New text related to runoff]**

In the Abstract, we now write: "We also provide an example assessment of a terrestrial impact: changes in mean runoff, and compare projections by the IMOGEN system which has one land surface model, versus direct GCM outputs which all have alternative representations of land functioning. The latter is noted as an additional source of uncertainty."

The main part of the paper, discussing the runoff findings, is repeated in full here, as: "In order to exemplify the ability of IMOGEN to project changes to impacts, we report results of the mean annual

total runoff ($R_{tot}$, mm day$^{-1}$) simulation based on the SRESA1B emissions forcing scenario (Figure 7), and compare them directly to GCM estimates of change in the same quantity. Hence this is comparing the IMOGEN simulations that emulate multiple GCMs but with a single land surface model (JULES), versus runoff values directly from the GCMs. The latter therefore contain alternative estimates of climate change, as well as the responses of different land surface models. During modelled pre-industrial control "spin-up" period, and modelled period centred on year 2090, total runoff values are recorded for each GCM (both emulated in IMOGEN, and directly from GCMs). Then in each case, the change is calculated. In the top panels of Figure 7, the mean of these changes are shown, whilst the bottom panels are the standard deviations of these change values. Although there are similarities between left and right-hand panels (over northern latitudes, in particular), there are important differences too, and notably the drying signal in GCM output for Africa and Australia is not reflected in the IMOGEN framework. For SDs, in some locations there is higher variability for IMOGEN than for the GCMs themselves; however, this pertains mainly to the regions where IMOGEN predicts higher runoff. The latter may be surprising, as considering that GCMs directly introduce another level of uncertainty i.e. inter-land surface model differences. Our finding is suggestive that JULES has a particularly sensitive response of runoff to imposed climatic changes. Runoff provides a challenge for comparison, as it is frequently a relatively small number between two larger fluxes of precipitation and evapotranspiration (transpiration, plus soil evaporation and interception loses) and so sensitive to change in those fluxes. Any direct comparison also needs to account for IMOGEN being initialised with a climatology based on the CRU dataset, and temporal dis-aggregation to sub-daily drivers of JULES having not been calibrated against any particular GCM. Nevertheless, to be a useful tool for impacts assessment, then IMOGEN must capture the general features of GCM projections of quantities such as runoff when operated for similar emissions scenarios. "

The component of the paper suggesting that direct scaling if impacts might be beneficial is written out in the manuscript as: "Looking ahead to further model development, one possibility is, for different GCMs, to pattern-scale directly impact variables of interest such as runoff against global land temperature change. This could be beneficial for two reasons. First, it would remove the current IMOGEN mismatch of many GCMs emulated based on their climate projections only, while the emulating system is coupled to just one land surface model. Instead, a more accurate representation would be gained of the spread of runoff uncertainty. Second, it would make model calculations computationally very fast, as full operation of the JULES system would not be required. Such an approach would be applicable when using IMOGEN to estimate changes for different future greenhouse gas concentrations, rather than land surface modelling development. A further possibility is to connect the meteorological pattern-scaling structure to alternative land surface models."

**[New References]**

The recent application of the IMOGEN framework to assess climate-permafrost thaw feedbacks is noted by citing: Burke, E.J., Ekici, A., Huang, Y., Chadburn, S.E., Huntingford, C., Ciais, P., Friedlingstein, P., Peng, S.S. and Krinner, G.: Quantifying uncertainties of permafrost carbon-climate feedbacks, Environ. Res. Lett., 14, 3051-3066, doi:10.5194/bg-14-3051-2017, 2017.

More recent information on the latest form of CRU data (CRU data provides the background climate to which anomalies are added) is now cited as: Harris, I., Jones, P.D., Osborn, T.J. and Lister, D.H.: Updated high-resolution grids of monthly climatic observations – the CRU TS3.10 Dataset., Int. J. Climatol., 34, 623-642, doi:10.1002/joc.3711, 2014.

In the Discussion, we now describe more extensively future enhancements to the pattern-scaling system. We write: "There are a number of potential methodological enhancements that can be implemented in the next IMOGEN version and beyond just fitting to the CMIP5 dataset (Taylor et al 2012). For example, so far the natural variability around the trend in IMOGEN is simulated through a daily "weather generator" component with invariant properties, and with no representation of inter-annual variability. However variability might also change in a warming world, and at a range of timescales from sub-daily through to major alteration at inter-annual timescales (e.g. Huntingford et al., 2013a). This suggests that in future research, at least for some variables, additional patterns might be added that capture such variability changes, and including any inter-annual variability and adjustment for different warming levels." Huntingford, C., Jones, P.D., Livina, V.N., Lenton, T.M. and Cox, P.M.: No increase in global temperature variability despite changing regional patterns, Nature, 500, 327-330, doi:10.1038/nature12310, 2013a.

IMOGEN has strong potential to be used for analysis of climate and terrestrial land surface implications of stabilization of warming at either 1.5°C or 2.0°C. It allows interpolation from GCMs to any new temperature scenarios. This makes the paper timely, given the new IPCC report specifically on such stabilization temperature limits. For this, we write in the paper: "The IMOGEN modelling system is available to determine future climate change, now with uncertainty, and forced by either a future pathway in either $CO_2$ emissions or $CO_2$ concentrations. A further and rapidly emerging application is to understand regional climate impacts during transition to different global thermal limits, with an emphasis on eventual stabilisation at 1.5°C or 2.0°C of global warming above pre-industrial levels. In this instance, most of the EBM in IMOGEN is overridden with a global temperature pathway (the land-ocean contrast and oceanic fraction cover only used to obtain $\Delta T_l$), but relying on the remaining spatial and monthly patterns to give detailed local climatic implications. It is planned to use different global temperature pathways to those two stabilised limits (Huntingford et al, 2017) to force IMOGEN in this configuration" Huntingford, C., Yang, H., Harper, A., Cox, P.M., Gedney, N., Burke, E.J., Lowe, J.A., Hayman, G., Collins, W.J., Smith, S.M. and Comyn-Platt, E.: Flexible parameter-sparse global temperature time profiles that stabilise at 1.5 and 2.0 degrees, Earth Syst. Dynam., 8, 617-626, doi:10.5194/esd-8-617-2017, 2017.

Related to the above is also a recent paper (James et al), offering the range of scientific approaches to 1.5°C or 2.0°C. Pattern-scaling is presented as a key tool. In the introduction, we write: "….potentially to investigate pre-defined future temperature thresholds such as two degrees of global warming above pre-industrial levels. Pattern-scaling has been suggested as a key methodology to understand the differences between climate stabilisation at either 1.5°C or 2.0°C (James et al. 2017)." James, R., Washington, R., Schleussner, C.-F., Rogelj, J., and Conway, D.: Characterizing half-a-degree difference: a review of methods for identifying regional climate responses to global warming targets, WIREs Clim. Change, 8, e457, doi:10.1002/wcc.457, 2017.

A recent overall assessment of pattern-scaling appeared in a far-reaching 2014 paper, which we now cite. Tebaldi, C and Arblaster, J.M.: Pattern scaling: Its strengths and limitations, and an update on the latest model simulations, Climatic Change, 122, 459-471, doi: 10.1007/s10584-013-1032-9, 2014.

**[Acknowledging CMIP5 & AR5]**. We are now very clear in the paper that the next step is IMOGEN with patterns from CMIP5. We write: "The [pattern] 
[revised manuscript text omitted]

---

## Author Response (AR4)

**CEH Wallingford**
Wallingford, OXON, OX10 8BB, U.K.
Email: chg@ceh.ac.uk
Tel: +44 (0)7884437138          Nov. 15th 2017

Prof. Wilco Hazeleger
Editor
Geoscientific Model Development

Dear Prof. Hazeleger,

Thank you for your continued support of our manuscript and the additional annotations on paper:

**"Climate patterns scaling set for an ensemble of 22 GCMs – adding uncertainty to the IMOGEN impacts system".**

We have answered these extra comments and requests, and by editing in "track changes" a version with previous adjustments all accepted. It has also resulted in four additional references being cited. This new version has been uploaded to the GMD website.

We hope that the manuscript is now near to being acceptable for publication in Geoscientific Model Development. Please do not hesitate to contact me if there are any further queries.

Thank you again for all of your help so far.

With kind regards,

Chris Huntingford

[revised manuscript text omitted]